# GPTailor: Large Language Model Pruning Through Layer Cutting and Stitching

**Guinan Su[1] , Li Shen[5]\* , Lu Yin[6] , Shiwei Liu[1,2,3] , Yanwu Yang[4] , Jonas Geiping[1,2,3]**

[1]Max Planck Institute for Intelligent Systems , [2]ELLIS Institute Tübingen
[3]Tübingen AI Center , [4]University of Tübingen,
[5] Shenzhen Campus of Sun Yat-sen University, [6]University of Surrey

## Abstract

Large language models (LLMs) have shown remarkable capabilities in language understanding and generation. However, such impressive capability typically comes with a substantial model size, which presents significant challenges in deployment and inference. While structured pruning of model parameters offers a promising way to reduce computational costs at deployment time, current methods primarily focus on single model pruning. In this work, we develop a novel strategy to compress models by strategically combining or merging layers from finetuned model variants, which preserves the original model's abilities by aggregating capabilities accentuated in different finetunes. We pose the optimal tailoring of these LLMs as a zero-order optimization problem, adopting a search space that supports three different operations: (1) Layer removal, (2) Layer selection from different candidate models, and (3) Layer merging. Our experiments demonstrate that this approach leads to competitive model pruning, for example, for the Llama2-13B model families, our compressed models maintain approximately 97.3% of the original performance while removing $\sim 25\%$ of parameters, significantly outperforming previous state-of-the-art methods. The code is available at.[1]

## 1 Introduction

The unique strengths of modern Large Language Models (LLMs) in language understanding, generation, and reasoning (Touvron et al., 2023; OpenAI et al., 2023; Chiang et al., 2023) are inextricably linked to their immense size. Research in this field has generally followed a trajectory of scaling model parameters and data to enhance performance, guided by two fundamental principles: scaling laws, which establish that performance improves predictably with increased parameters (Kaplan et al., 2020; Hoffmann et al., 2022; Wei et al., 2022), and over-parameterization theory, which demonstrates that models with excess parameters achieve better optimization and generalization (Allen-Zhu et al., 2019a;b; Li et al., 2020). These principles have led researchers to develop billion-parameter architectures delivering unprecedented performance across diverse language tasks.

Despite these impressive capabilities, deploying LLMs presents significant challenges due to their substantial computational demands. Various post-training techniques have been proposed to address the issues faced when deploying models to consumer GPUs or local devices, or when reducing costs, including model pruning (Frantar & Alistarh, 2023; Dettmers et al., 2023b; Xia et al., 2023; Kim et al., 2024; Ma et al., 2023), knowledge distillation into smaller models (Chen et al., 2022; Hsieh et al., 2023; Shridhar et al., 2023; Tunstall et al., 2023), and quantization of weights (Yao et al., 2022; Gholami et al., 2022; Dettmers et al., 2023a). While quantization reduces parameter precision but requires specific hardware support, and knowledge distillation necessitates costly retraining of smaller models, structured pruning offers a more flexible and hardware-agnostic approach by eliminating redundant parameters to decrease computation costs.

---

\*Correspondence to: shenli6@mail.sysu.edu.cn
[1]https://github.com/Guinan-Su/auto-merge-llm

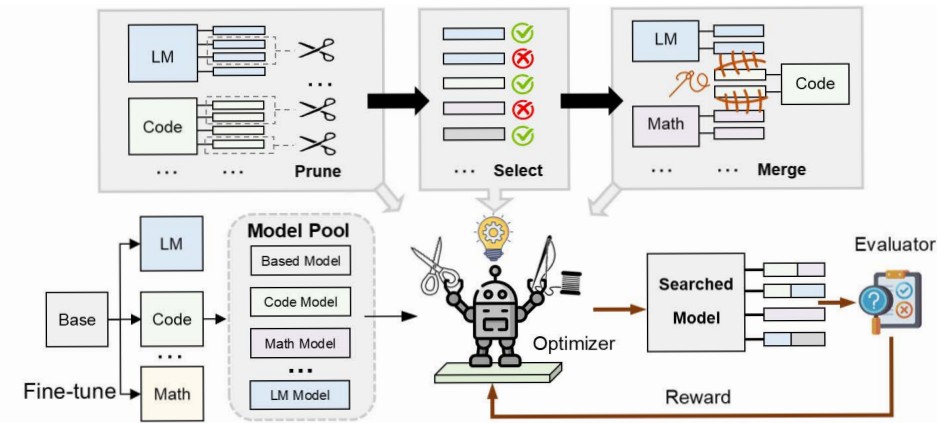

Figure 1: Our Approach: **Model Pruning through Cutting and Stitching**. We achieve competitive model pruning performance by running a zero-order search that tailors layers based on a shared pool of finetuned variants of the original model, selecting and stitching layers if necessary. The model finetunes accentuate task-specific skills, allowing us to merge key components into a smaller model, maintaining, for example, 97% of the capabilities of Llama-13B, even after a 25% reduction in layers.

Existing pruning methods typically focus on pruning individual models through manually designing metrics that assess the importance of specific structures or layers based on hidden state changes or gradient (Kim et al., 2024; Men et al., 2024; Ma et al., 2023). However, most of these approaches cause performance degradation and require additional post-training with full parameters to recover performance.

To address these limitations, we take a radically different perspective and re-formulate structured pruning as the problem of *pruning not individual models, but a family of task-specific finetuned versions of a given model*. These finetuned variants are surprisingly helpful for model pruning, as each variant accentuates a particular task, such as coding, math, or language understanding. Further, the variants are close enough that model merging can be employed to re-combine layers from multiple variants, if needed (Wortsman et al., 2022). These observations lead us to our main question: **Can we develop better compressed models by strategically combining or merging layers from different models?** Motivated by this question, we propose a novel structured pruning method based on zero-order optimization that supports three different operations to combine layers from different models into a smaller, more efficient model: **(1) Layer removal, (2) Layer selection from related candidate models, (3) Layer merging**.

For the optimization, we define multiple objective functions that capture different aspects of model performance across different tasks to better preserve the original model's capabilities and run a fully data-driven zero-order optimization, instead of relying on expert-made heuristics for pruning. We employ SMAC (Lindauer et al., 2022), which strategically allocates computational resources by evaluating configurations at different calibration data sizes, thereby reducing computational costs while boosting the efficiency of finding superior solutions. We rigorously validate our method's effectiveness by evaluating it on Llama-7B and Llama-13B with four state-of-the-art structural pruning methods across comprehensive benchmarks. Our experimental results demonstrate that our approach maintains excellent performance while outperforming existing pruning methods.

In summary, the main contributions of this paper are:

- We propose a novel structured pruning method that formulates pruning as a zero-order optimization problem over a pool of candidate models, enabling automated discovery of efficient models that leverage capabilities from multiple models.
- We find that this approach allows for a cost-effective model pruning stage that is effective without the need for post-training to heal the pruned model.
- We validate our method's effectiveness through extensive experiments, comparing against modern LLM pruning methods on 14 benchmark tasks.

Our method maximally preserves the capabilities of the dense model: 92.2% for the 7B model and 97.3% for the 13B model. significantly outperforming previous state-of-the-art methods.

## 2 RELATED WORK

**Compression of Language Models.** Large language models (Touvron et al., 2023; OpenAI et al., 2023; Chiang et al., 2023) necessitate efficient compression methods to reduce parameters and latency. These methods include structural pruning (Frantar & Alistarh, 2023; Dettmers et al., 2023b; Xia et al., 2023; Kim et al., 2024; Ma et al., 2023), knowledge distillation (Chen et al., 2022; Hsieh et al., 2023; Tunstall et al., 2023), and quantization (Yao et al., 2022; Gholami et al., 2022; Dettmers et al., 2023a). Our work focuses on structural pruning, which removes sub-components from neural networks for hardware-friendly compression - instead of pruning through sparsification, which requires significant effort to materialize gains on standard hardware. Recent pruning methods are typically guided by expert-designed criteria. LLMPruner (Ma et al., 2023) removes non-critical structures using gradient information. SliceGPT (Ashkboos et al., 2024) reduces dimensionality by replacing weight matrices with smaller ones. LaCo (Yang et al., 2024) collapses the weights of later layers into earlier ones based on activation similarity. ShortGPT (Men et al., 2024) measures layer importance through Block Influence (BI) derived from hidden state changes. Unlike these metric-based methods targeting individual models, our approach employs zero-order *search*, namely hyperparameter optimization, to combine pruning and merging across model families. While LaCo also uses layer merging, it focuses only on merging similar layers for a single model, whereas we focus on strategically combining or merging layers from different models, which we find to noticeably improve upon within-model merging. Additionally, our approach differs from the weight-sharing NAS-based pruning method (Klein et al., 2024), which requires costly training. Instead of searching within a single model, we directly optimize across fine-tuned models, strategically combining layers from diverse variants.

**Model Merging.** Model merging enhances capabilities without additional training data or computation(Yang et al., 2026; Li et al., 2023b). The field evolved from simple weighted parameter averaging (Utans, 1996) that often yielded suboptimal results to advanced techniques like Task Arithmetic (Ilharco et al., 2022) which computes differences between model parameters, and SLERP (White, 2016) which performs interpolation along spherical paths. Later approaches leveraged neural network sparsity, with TIES-Merging (Yadav et al., 2024) selecting parameters based on magnitude while addressing sign conflicts, and DARE (Yu et al., 2024) combining sparsification with parameter rescaling. AdaMerging(Yang et al., 2023) introduces adaptive merging coefficients learned in both task and layer-wise manners, while WEMoE (Shen et al., 2025) constructs a weight-ensembling mixture of experts from merged models. Further developments model the merging process as adaptive projective gradient descent (Wei et al., 2025) and address scalable continual model merging through sequential approaches that avoid retraining (Tang et al., 2025). Recent advances include Evolutionary model merging (Akiba et al., 2024), optimizing coefficients through evolutionary search, and multi-fidelity approach (Su & Geiping, 2025) that enables fine-grained exploration while reducing costs. Our work also builds upon a multi-fidelity optimization framework to allow for an efficient search for compressed models.

## 3 METHODS

In this section, we provide a detailed explanation of our approach. Unlike conventional model compression pipelines, we formulate pruning as a zero-order optimization problem over the layers and merging hyperparameters of a set of candidate models. We begin in Section 3.1 by outlining our problem formulation and defining the optimization pipeline for pruning with three key components: a search space, a target objective, and an optimizer. Section 3.2 follows with a description of the search spaces. In Section 3.3, we introduce our designed target objective function. Finally, In Section 3.4, we describe our choice of optimization strategy, which efficiently navigates the defined search space to identify optimal pruning configurations. An overview of the pipeline is provided in Figure 1.

### 3.1 PROBLEM SETUP

Given a pre-trained base model $M_{\text{base}}$ and a set of candidate models $\mathcal{M} = \{M_1, M_2, ..., M_K\}$ fine-tuned from the same base model, our goal is to find an optimal pruned model that maximizes performance while adhering to a target sparsity constraint. Let $s$ denote the target sparsity factor, where $s \in [0, 1]$ indicates the fraction of parameters to be pruned. The pruned model is constructed through a combination of layers from candidate models, employing operations such as layer-wise

merge, layer selection, and layer removal. These combinations and operations are determined by a set of hyperparameters $\omega \in \Omega$, with $\Omega$ representing the search space of all possible hyperparameter configurations. Each configuration $\omega$ defines a specific way to combine the layers from candidate models to form a pruned model $M_\omega$. The performance of the pruned model can be evaluated using a function $f(M_\omega)$, which measures the model's effectiveness on specific datasets and tasks. This leads to our optimization problem:

$$\omega^* = \arg\min_{\omega \in \Omega} f(M_\omega) \quad \text{subject to} \quad \text{S}(M_\omega) \leq s \tag{1}$$

where $\text{S}(\cdot)$ calculates the fraction of pruned parameters in the model compared to the base model, and $\omega^*$ represents the optimal hyperparameter configuration that yields the performing pruned model.

## 3.2 SEARCH SPACE DESIGN

The search space $\Omega$ encompasses all possible pruning configurations that can be applied to construct our pruned model. We formulate this space based on structural layer-wise pruning operations. We aim to support three operations: (1) Layer removal, (2) layer selection, and (3) Layer merging. We designed our search space as follows:

Given a base model with $l$ layers and $K$ candidate models fine-tuned from this base model, we design the search space through a binary vector $\mathbf{r} = [r_1, r_2, \ldots, r_l]$ where $r_i \in \{0, 1\}$ indicates whether the $i$-th layer is retained ($r_i = 0$) or removed ($r_i = 1$), satisfying $\sum_{i=1}^{l} r_i = \lceil l \cdot s \rceil$ to achieve our target sparsity $s$. For each retained layer position $i$, we define a selection vector $\mathbf{c}_i = [c_{i,1}, c_{i,2}, \ldots, c_{i,K}]$ where $c_{i,j} \in \{0, 1\}$ indicates whether the layer from the $j$-th candidate model is selected. If $\sum_{j=1}^{K} c_{i,j} = 0$, we retain the layer from the base model instead. When multiple candidate models contribute to a layer position (i.e., $\sum_{j=1}^{K} c_{i,j} > 1$), we specify a merge method $m_i \in \{1, 2, \ldots, Z\}$ from $Z$ available merging techniques. Each merge method $m_i$ is associated with a set of hyperparameters $\mathbf{h}_i = [h_{i,1}, h_{i,2}, \ldots, h_{i,P_i}]$, where $P_i$ is the number of hyperparameters for the specific merge method. These hyperparameters govern the precise mechanism of layer combination, such as interpolation weights or mask ratio parameters. Therefore, a complete configuration $\omega \in \Omega$ is represented as $\omega = \{\mathbf{r}, \{\mathbf{c}_i | r_i = 0\}, \{m_i | r_i = 0 \text{ and } \sum_{j=1}^{K} c_{i,j} > 1\}, \{\mathbf{h}_i | r_i = 0 \text{ and } \sum_{j=1}^{K} c_{i,j} > 1\}\}$. The total cardinality of the search space can be calculated as: $|\Omega| = \binom{l}{\lceil l \cdot s \rceil} \times \prod_{i:r_i=0} 2^K \times \prod_{i:r_i=0, \sum_{j=1}^{K} c_{i,j} > 1} Z \times \prod_{i:r_i=0, \sum_{j=1}^{K} c_{i,j} > 1} |\mathbf{h}_i|$. which enables a wide exploration of pruning strategies while maintaining the target sparsity constraint.

## 3.3 TARGET OBJECTIVE FUNCTION

To evaluate the quality of a pruned model, we define a multi-objective function that measures the model's effectiveness across tasks. Specifically, we measure performance on calibration datasets $\mathcal{D}_{\text{calibration}}$, quantifying metrics such as accuracy for classification tasks or perplexity for language modeling tasks. This provides a direct assessment of how well the pruned model preserves the capabilities of the original model. We define a multi-task objective function that captures different aspects of model performance across a range of tasks to produce a comprehensive pruned model. Let $\mathcal{T} = \{T_1, T_2, \ldots, T_m\}$ be a set of $m$ tasks. For a pruned model $M_\omega$ with configuration $\omega$, we employ Pareto Efficient Global Optimization (ParEGO) (Knowles, 2006) to identify Pareto-optimal solutions across different objectives. Specifically, the ParEGO algorithm transforms multi-objective optimization problems into a series of single-objective problems through scalarization methods:

$$f_{\text{multi}}(M_\omega, \lambda) = \max_{i=1,\ldots,m} \{\lambda_i \cdot f_i(M_\omega)\} + \alpha \sum_{i=1}^{m} \lambda_i \cdot f_i(M_\omega) \tag{2}$$

where $f_i(M_\omega)$ is the $i$-th objective function, $\lambda_i$ is the corresponding weight satisfying $\sum_{i=1}^{m} \lambda_i = 1$ and $\lambda_i \geq 0$, and $\alpha$ is a small positive constant (typically set to 0.05). The Chebyshev norm component $\max_{i=1,\ldots,m}\{\lambda_i \cdot f_i(M_\omega)\}$ ensures that all non-dominated solutions on the non-convex Pareto front can be identified, while the term $\alpha \sum_{i=1}^{m} \lambda_i \cdot f_i(M_\omega)$ enhances the algorithm's stability. The final output of our optimizer is a Pareto front of pruning configurations, where each configuration represents

a different trade-off between performance on various tasks. In our experiments, we selected the configurations from the best performing Pareto front and report their results.

## 3.4 Search Optimizer

To efficiently navigate the search space and find optimal pruning configurations, we employ SMAC (Lindauer et al., 2022), which strategically allocates computational resources by evaluating configurations at different fidelity levels. we use calibration dataset size as fidelity type, represented by budgets b where $b_{\min} \leq b \leq b_{\max}$. Each budget value corresponds to a specific portion of the calibration data used for evaluation - smaller budgets (lower fidelity) use fewer samples for faster but less precise evaluations, while larger budgets (higher fidelity) use more samples for slower but more accurate assessments. We use Random Forest (Breiman, 2001) as a surrogate model to sample new configurations. Given configuration space $\Omega$, minimum budget $b_{\min}$, maximum budget $b_{\max}$, reduction factor $\eta$ and the maximum trials $T_{\max}$, the whole process is described in Algorithm 1.

---

**Algorithm 1** The optimization process of Gptailor.

---

**Require:** Configuration space $\Omega$, minimum budget $b_{\min}$, maximum budget $b_{\max}$, reduction factor $\eta$, maximum trials $T_{\max}$
**Ensure:** Optimized configuration $\omega^*$
1: $s_{\max} = \lfloor \log_\eta \frac{b_{\max}}{b_{\min}} \rfloor$, $D \leftarrow \emptyset$, $T \leftarrow 0$                 ▷ Initialization
2: **for** $s \in \{s_{\max}, s_{\max} - 1, \ldots, 0\}$ and $T < T_{\max}$ **do**
3:      $n \leftarrow \lceil \frac{(s_{\max}+1)}{(s+1)} \cdot \eta^s \rceil$, $r \leftarrow b_{\min} \cdot \eta^s$          ▷ Config count & budget
4:      $\mathcal{C} \leftarrow$ Sample Configurations($n, D, \Omega$)         ▷ Sample configurations
5:      **for** $i \in \{0, 1, \ldots, s\}$ and $T < T_{\max}$ **do**
6:          $n_i \leftarrow \lfloor n \cdot \eta^{-i} \rfloor$, $r_i \leftarrow r \cdot \eta^i$          ▷ Stage parameters
7:          **for** each $w \in \mathcal{C}$ and $T < T_{\max}$ **do**
8:              Evaluate $y_w \leftarrow f_{\text{multi}}(M_w, \lambda)$ using $r_i$ samples from calibration set, $D \leftarrow D \cup \{(w, r_i, y_w)\}$, $T \leftarrow T + 1$
9:          **end for**
10:          Sort $\mathcal{C}$ by performance, keep the top $\lfloor n_i/\eta \rfloor$ configurations in $\mathcal{C}$
11:      **end for**
12: **end for**
13: **return** the best-performing configuration $\omega^*$ evaluated at highest budget

---

This efficient optimization strategy enables us to handle the search space defined in Section 3.2, identifying high-performing pruned models that satisfy our multi-objective function from Section 3.3, with significantly reduced computational cost compared to exhaustive search approaches.

## 4 Experimental Settings

**Benchmarks.** To evaluate the pruned model's capabilities, we utilized the OpenCompass evaluation framework (Contributors, 2023). Specifically, we conduct evaluations in five aspects: Reasoning, Language, Knowledge, Examination and Understanding. Reasoning: CMNLI (CNLI)(Xu et al., 2020), HellaSwag (HeSw)(Zellers et al., 2019), PIQA (Bisk et al., 2020). Language: CHID (Zheng et al., 2019), WSC (Levesque et al., 2012). Knowledge: CommonSenseQA (CSQA) (Talmor et al., 2018), BoolQ (Clark et al., 2019). Examination: MMLU (Hendrycks et al., 2020), CMMLU (CMLU) (Li et al., 2023a). Understanding: Race-High/Middle (H/M) (Lai et al., 2017), XSum (Narayan et al., 2018), C3 (Sun et al., 2020). For CHID and XSum, we use generative evaluation. For the WSC dataset, we use cloze log-likelihood (WSCP) and generative (WSCG) evaluation. The remaining benchmarks are evaluated using cloze log-likelihood. See more details in Supplementary Section C.

**Baselines.** To evaluate the effectiveness of our method, we compared with four state-of-the-art structured pruning methods: LLM-Pruner (LLMPru) (Ma et al., 2023), SliceGPT (Ashkboos et al., 2024), LaCo (Yang et al., 2024), and ShortGPT (Men et al., 2024). In our experiments, we set the pruning ratios of our method to be equivalent to ShortGPT and LaCo. Furthermore, as our method is based on multiple candidate models, we check three comprehensive comparison scenarios to guarantee fairness: (1) Applying each baseline pruning method individually to all candidate models and picking

the strongest one, (2) First pruning each candidate model using the baseline method and then merging them, and (3) First merging the candidate models and then applying pruning. For model merging across baseline experiments, we employ the task-arithmetic merging (Ilharco et al., 2022) technique used in our search space, with merging factors within the range [0.5, 1.0] (Ilharco et al., 2022).

**Model Selection.** To assess the effectiveness of the proposed method, we search for pruned versions of the popular Llama2-7B and Llama2-13B (Touvron et al., 2023). For 7B models, we use Llama-2-7B (Touvron et al., 2023) as our base model, with three candidate models: Llama-2-7B-Chat (Touvron et al., 2023) (LM), MAmmoTH-7B (Yue et al., 2023) (Math), and Llama-2-Coder-7B (Manuel Romero, 2023) (Code). For 13B models, we use Llama-2-13B (Touvron et al., 2023) as the base model, with WizardLM-13B (Xu et al., 2023) (LM), WizardMath-13B (Luo et al., 2023) (Math), and Llama-2-13B-Code-Alpaca (Chaudhary, 2023) (Code) as candidate models. We selected these models for their wide availability to ensure reproducible evaluation. For the 7B models, we set the sparsity ratio to 9/32, removing approximately 28% of the layers. For the 13B models, we set the sparsity ratio to 10/40, removing approximately 25% of the layers. These two ratios are matching the best settings from prior work in ShortGPT and LaCo, while being slightly higher than other baseline methods, allowing for fair comparisons. For layer merging, we implement task-arithmetic (Ilharco et al., 2022) merging with a configurable merging factor that controls the magnitude of task-specific adaptations.

**Calibration Data.** For our calibration dataset, we selected multiple-choice datasets to ensure the model's generalization ability across different capabilities. Specifically, we sampled from diverse datasets: 1000 examples from the PIQA (Bisk et al., 2020) training set, 500 examples from the WSC (Levesque et al., 2012) training set, 1000 examples from the CSQA Talmor et al. (2018) training set, and 1000 examples from the MMLU (Hendrycks et al., 2020) validation set (which is distinct from the MMLU test set). This diverse collection allows us to calibrate our model across a broad spectrum of linguistic and reasoning capabilities.

**Objective and Optimizer.** Our implementation builds upon SMAC (Lindauer et al., 2022) for optimization. We allocate 500 search trials for both 13B and 7B experiments. To improve optimization efficiency, we use models with randomly removed middle layers as starting points, since models are relatively robust to changes in these intermediate layers (Su & Geiping, 2025). We set the minimum budget $b_{min}$ as 100, maximum budget $b_{max}$ as the 1000, and reduction factor $\eta$ as 3. This resulted in budgets of {100, 300, 1000} for PIQA, CSQA, and MMLU. For the WSC, we set budgets to {100, 200, 500}

## 5 RESULTS AND ANALYSIS

### 5.1 MAIN RESULTS

To validate the effectiveness of our method, we compared it with the four baselines: LLM-Pruner (LLMPru) (Ma et al., 2023), SliceGPT (Ashkboos et al., 2024), LaCo (Yang et al., 2024), and ShortGPT (Men et al., 2024). We reproduce the results from these methods and evaluate on Open-Compass (Contributors, 2023). As mentioned in the experiment section, we evaluate the results based on three settings, i.e., individual pruning, pruning-then-merging, and merging-then-pruning.

Table 1 reports the best single model pruning and best merge results of all baselines, with full results in Supplementary Section G. Our approach achieves the best results across multiple benchmarks compared to all tested LLM pruning methods. In terms of overall performance, our method maximally preserves the capabilities of the dense model: 92.2% (48.55/52.63) for the 7B model and 97.3% (54.33/55.86) for the 13B model. To ensure our results were not biased by our calibration data, we also calculate an avg* excluding the four benchmarks from which training data was selected for calibration (MMLU, CSQA, WSC, PIQA). As shown in the avg* column, our method still outperformed all baselines, further validating our approach. Notably, our method achieved comparable or even better results than dense models on most tasks. We attribute these gains to: 1) Pruning might mitigate "overthinking" effects (Kaya et al., 2019), as evidenced by benchmarks such as CNLI and WSC, where other pruning methods also yielded performance gains, and 2) Our merging strategy might mitigate the information loss caused by pruning, stemming from the merging process.

Figure 2 illustrates our best-performing 7B-pruned model and best-performing 13B-pruned models' structure (See Supplementray Table 12 and Table 13 for details). We observe that both models tend to remove middle-to-later layers, with the 13B model removing layers from layer 25 and the 7B model

Table 1: Comparison of pruning methods on multiple natural language benchmarks. "Single" refers to the performance achieved when pruning a single model directly, while "Merge" refers to the performance achieved through either "pruning-then-merging" or "merging-then-pruning". 7B models: Llama-2-7B-Chat (LM), MAmmoTH-7B (Math), Llama-2-Coder-7B (Code), and Llama-2-7B (Base). 13B models: WizardLM-13B (LM), WizardMath-13B (Math), llama-2-13B-code-alpaca (Code), and Llama-2-13B (Base).

| LLM | Pruner (ratio) | Type | Reasoning CNLI | HeSw | PIQA | Language CHID | WSC$_P$ | WSC$_G$ | Knowledge CSQA | BoolQ | MMLU | CMLU | Understanding Race$_H$ | Race$_M$ | XSum | C3 | Avg | Avg* |
|---|---|---|---|---|---|---|---|---|---|---|---|---|---|---|---|---|---|---|
| Llama-7B | Dense (0.0%) | Base | 32.98 | 71.34 | 78.18 | 41.56 | 37.50 | 38.46 | 55.04 | 70.70 | 46.67 | 31.88 | 35.53 | 33.36 | 19.55 | 43.84 | 45.47 | 42.30 |
| | | Math | 32.99 | 68.60 | 75.79 | 39.71 | 39.42 | 36.54 | 50.78 | 69.36 | 43.04 | 32.16 | 30.36 | 36.42 | 20.88 | 43.45 | 44.25 | 41.70 |
| | | LM | 31.30 | 71.28 | 75.95 | 36.11 | 63.46 | 59.62 | 64.29 | 74.77 | 48.30 | 33.93 | 52.52 | 55.22 | 22.45 | 47.56 | 52.63 | 47.24 |
| | | Code | 32.99 | 70.27 | 78.62 | 41.61 | 36.54 | 41.35 | 57.41 | 71.04 | 46.22 | 32.20 | 41.25 | 39.69 | 18.79 | 46.25 | 46.73 | 43.79 |
| | LLMPru (25.3%) | Single | 32.99 | **59.57** | **73.34** | 30.32 | 46.15 | 0.00 | 20.15 | 57.28 | 23.21 | 25.16 | 21.56 | 21.52 | **15.19** | 31.07 | 32.68 | 32.74 |
| | | Merge | 34.71 | **60.57** | 73.50 | 26.62 | 40.38 | 5.77 | 19.90 | 52.14 | 24.01 | 25.30 | 23.07 | 22.98 | **15.51** | 32.49 | 32.64 | 32.60 |
| | SliceGPT (26.3%) | Single | 31.89 | 41.55 | 58.81 | 18.43 | 39.42 | 4.81 | 19.49 | 40.09 | 25.38 | 25.02 | 25.59 | 26.88 | 8.78 | 39.56 | 28.98 | 28.64 |
| | | Merge | 32.85 | 37.61 | 57.56 | 17.33 | 53.85 | 2.88 | 19.41 | 42.66 | 25.22 | 24.68 | 25.21 | 24.72 | 12.78 | 40.22 | 29.78 | 28.67 |
| | LaCo (27.1%) | Single | 32.97 | 55.24 | 69.53 | **31.47** | 36.54 | 34.62 | 22.11 | 67.22 | 29.08 | 26.16 | 28.53 | 28.27 | 14.68 | **43.51** | 37.14 | 36.45 |
| | | Merge | 31.89 | 56.26 | **71.22** | 27.32 | 39.42 | 22.12 | 23.42 | 72.66 | 29.30 | 26.00 | 25.19 | 26.81 | **16.11** | **43.62** | 36.52 | 36.21 |
| | ShortGPT (27.1%) | Single | 33.09 | 57.42 | 66.54 | 21.53 | 56.73 | **48.08** | 52.50 | 67.34 | 43.68 | 28.31 | 32.53 | 31.69 | 12.40 | 39.45 | 42.24 | 35.97 |
| | | Merge | 34.10 | 54.18 | 64.42 | 16.83 | 61.54 | 36.54 | 55.61 | **73.21** | 36.84 | 25.61 | 42.94 | 45.89 | 10.12 | 35.73 | 42.40 | 37.62 |
| | Ours (27.1%) | | **35.46** | 54.43 | 67.74 | 23.63 | **63.46** | 43.27 | **62.90** | 75.08 | 48.75 | 33.86 | 55.35 | 58.64 | 12.99 | **44.16** | 48.55 | 43.73 |
| Llama-13B | Dense (0.0%) | Base | 32.99 | 74.77 | 79.71 | 47.35 | 50.96 | 63.46 | 67.24 | 71.38 | 55.84 | 38.74 | 57.98 | 60.17 | 23.47 | 47.51 | 55.11 | 50.48 |
| | | LM | 35.36 | 70.41 | 78.73 | 36.21 | 57.69 | 60.58 | 65.03 | 73.70 | 53.48 | 30.85 | 66.12 | 71.66 | 22.44 | 52.00 | 55.30 | 50.97 |
| | | Math | 32.99 | 68.78 | 77.26 | 44.36 | 36.54 | 19.23 | 60.36 | 78.44 | 54.21 | 38.12 | 47.74 | 48.82 | 19.51 | 44.66 | 47.93 | 47.05 |
| | | Code | 32.99 | 74.82 | 80.14 | 47.30 | 51.92 | 63.46 | 68.88 | 72.72 | 55.92 | 39.26 | 58.03 | 63.72 | 24.45 | 48.38 | 55.86 | 51.30 |
| | LLMPru (21.2%) | Single | **33.49** | 60.28 | **75.57** | 23.68 | 39.42 | 0.00 | 19.00 | 63.24 | 23.27 | 25.23 | 22.36 | 21.45 | **17.13** | 32.00 | 32.58 | 33.21 |
| | | Merge | **33.86** | 64.11 | 73.50 | 22.18 | **60.58** | 0.00 | 21.46 | 61.96 | 23.84 | 25.62 | 22.16 | 21.59 | 14.98 | 32.11 | 34.14 | 33.17 |
| | SliceGPT (23.6%) | Single | 33.19 | 42.44 | 59.90 | 18.03 | 54.81 | 19.23 | 32.51 | 41.22 | 33.09 | 25.75 | 29.45 | 29.87 | 9.99 | 37.75 | 33.37 | 29.74 |
| | | Merge | 30.98 | 46.83 | 62.57 | 19.33 | 51.92 | 49.04 | 37.76 | 38.38 | 33.55 | 25.22 | 23.53 | 23.05 | 9.95 | 39.67 | 35.13 | 28.55 |
| | LaCo (24.6%) | Single | 32.33 | 60.18 | 70.57 | **32.67** | 34.62 | 34.62 | 52.58 | 62.66 | 36.26 | 25.80 | 60.38 | 62.53 | 8.79 | **49.21** | 44.51 | 43.84 |
| | | Merge | **33.49** | 62.50 | 74.37 | **35.26** | **63.46** | 63.46 | 18.84 | 64.65 | 41.83 | 24.87 | 26.10 | 25.97 | 15.93 | 39.51 | 42.16 | 34.71 |
| | ShortGPT (24.6%) | Single | 32.95 | 62.64 | **73.50** | 28.22 | 36.54 | 50.96 | 65.44 | 67.71 | 53.50 | 30.73 | **65.52** | 71.38 | **19.12** | **48.60** | 50.49 | 47.43 |
| | | Merge | 31.07 | 63.24 | 68.61 | 27.17 | 49.04 | 43.27 | 65.68 | **78.01** | 51.26 | 36.88 | 57.38 | 62.67 | **16.94** | 44.05 | 49.66 | 46.38 |
| | Ours (24.6%) | | 32.99 | **66.81** | 75.03 | 29.07 | 54.81 | 62.50 | **69.37** | 74.28 | 55.90 | 39.71 | 65.52 | 71.03 | 16.80 | 46.74 | 54.33 | 49.22 |

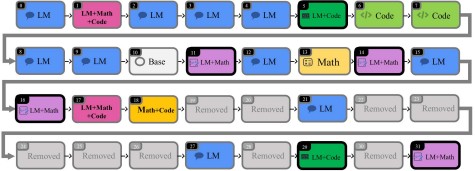
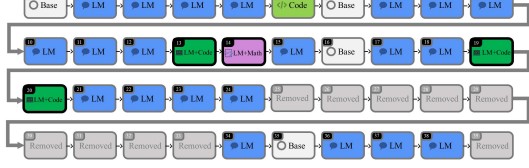

a) Structure of the best 7B-pruned model.          b) Structure of the best 13B-pruned model.

Figure 2: (a) Structure of our best-performing 7B-pruned model. The model integrates layers from multiple candidates: Llama-2-7B-Chat (LM), MAmmoTH-7B (Math), Llama-2-Coder-7B (Code), and Llama-2-7B (Base). The pruning ratio is 9/32, removing 9 layers out of 32 total layers. (b) Structure of our best-performing 13B-pruned model. The model integrates layers from multiple candidates: WizardLM-13B (LM), WizardMath-13B (Math), llama-2-13b-code-alpaca (Code), and Llama-2-13B (Base). The pruning ratio is 10/40, removing 10 layers out of 40 total layers.

from layer 19. This suggests information redundancy in these layers, aligning with findings that later layers exhibit high similarity and redundancy (Men et al., 2024; Gromov et al., 2024).

## 5.2    WHICH PARTS OF THE SEARCH SPACE ARE CRITICAL ?

To determine where the benefits of our approach come from, we designed ablation experiments to evaluate the contribution of different components in our search space. As our framework supports: (1) Layer Selection (LS) from different candidate models, (2) layer merging, and (3) Layer Removal

(LR), we conducted ablation studies to isolate the impact of each component. Table 2 summarizes the performance comparison across various benchmarks (More results in Supplementary Table 9).

**Layer Removal Only (LR-only).** We restricted the search space to allow only layer removal operations on a single model. Consequently, our method in this setting supports only single-model pruning without merging, similar to most conventional pruning approaches. As shown in Table 2, there is a significant performance drop (48.55% vs. 44.83%), confirming that merely pruning layers from a single model is insufficient. Moreover, it is worth noting that even with layer-removal only pruning on a single model, our method still outperforms the best baseline, ShortGPT (44.83% vs. 42.24%). This highlights the superiority of our approach to pruning, even in a simplified setting.

**Layer Selection and Removal (LS+LR).** In this setting, we enabled both layer selection from different candidate models and layer removal operations, while disabling the layer merging functionality. Compared with LR-only, LS+LR yields an even larger performance drop (48.55 vs. 43.20 on average). This suggests that merely combining layers from different models without proper integration through merging is ineffective.

Table 2: Comparison of different searching settings across various benchmarks. Settings: LR-only: Layer-remove only, LS+LR: Layer-selection + layer-remove, FL-merge: Folding Layers Merging.

| Setting | Reasoning | | | Language | | | Knowledge | | | | Understanding | | | | Avg |
|---|---|---|---|---|---|---|---|---|---|---|---|---|---|---|---|
| | CNLI | HeSw | PIQA | CHID | WSC$_P$ | WSC$_G$ | CSQA | BoolQ | MMLU | CMLU | Race$_H$ | Race$_M$ | XSum | C3 | |
| Ours | 35.46 | 54.43 | 67.74 | 23.63 | 63.46 | 43.27 | 62.90 | 75.08 | 48.75 | 33.86 | 55.35 | 58.64 | 12.99 | 44.16 | 48.55 |
| LR-only | 34.96 | 53.80 | 66.70 | 18.58 | 49.04 | 58.65 | 60.61 | 68.87 | 47.85 | 33.54 | 42.51 | 43.04 | 8.05 | 41.42 | 44.83 |
| LS+LR | 32.92 | 55.84 | 65.07 | 17.98 | 63.46 | 26.92 | 58.97 | 51.22 | 48.97 | 34.61 | 48.68 | 49.44 | 8.33 | 42.41 | 43.20 |
| FL-merge | 32.99 | 52.90 | 63.66 | 19.28 | 46.15 | 62.50 | 60.52 | 75.20 | 48.30 | 34.33 | 50.77 | 55.29 | 6.39 | 39.40 | 46.26 |

## 5.3 ROBUSTNESS AND GENERALIZATION ANALYSIS

To comprehensively evaluate the robustness and generalizability of our framework, we conduct extensive analysis across three critical dimensions: pruning ratio sensitivity, candidate pool scalability, and Generalization to next-generation models. These experiments aim to validate our method's effectiveness under diverse deployment constraints and resource limitations.

Table 3: Impact of Candidate Pool Composition on Performance.

| Model Pool | Average Performance |
|---|---|
| Math&LM&Code | **48.55** |
| Math&LM | 47.82 |
| Code&LM | 47.31 |
| Code&Math | 43.12 |
| LM | 45.42 |
| Math | 42.40 |
| Code | 42.03 |
| Base | 42.20 |

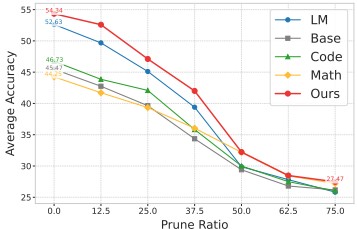

Figure 3: Performance Comparison Across Different Pruning Ratios.

**Performance Across Pruning Ratios.** To further evaluate the generalizability of our method under different pruning ratios, we validate its performance across varying pruning ratios. Since we have already shown that even the layer-removal variant of our method surpasses other baselines such as ShortGPT, here we focus specifically on layer removal. Moreover, we examine its impact on different task-specific models, using this experiment to highlight the additional benefits of merging, rather than simply pruning a single model. The results are visualized in Figure 3 with the average accuracy among benchmark performances at different pruning ratios. More details are supplied in Supplementary Table 10. From the results, we can see that the accuracy of all models decreases as the pruning ratio increases. Our model achieves the best performance at all pruning ratios, especially in the low pruning ratio range of 0%-37.5%. When pruning reaches 50%, every model suffers performance collapse, leading to a reduced gap across models. This represents a clear elbow point, indicating that beyond it, excessive parameter removal renders models unable to sustain effective functionality without further post-training.

**Scaling with Candidate Model Pool Size.** To validate the generalizability of our method across different candidate models, we conducted experiments by varying both the number of models and

their combinations in the pool. As shown in Table 3, with full results in Supplementary Table 21 the results show that performance is indeed affected by the choice of candidate models. Specifically, including language models (LM) in the candidate pool consistently yields substantial improvements, while code models tend to contribute more modest gains. Importantly, we find that increasing the number of candidate models consistently leads to improved overall performance. Our findings highlight three key properties of the proposed method. (1) Incorporating high-quality models, such as strong language models, consistently improves performance across benchmarks. (2) Adding lower-performing models does not harm the overall results, demonstrating the stability of our search strategy. (3) Enlarging the candidate pool generally yields further improvements, reflecting the scalability and robustness of our approach.

**Generalization to Next-Generation Models (Llama-3)**. We further extend our validation to Meta's Llama-3 8B model (Grattafiori et al., 2024), which is larger, more densely parameterized, and trained on 15T tokens with architectural improvements such as universal GQA and a longer context window. Despite a similar model size, Llama-3 8B surpasses Llama-2 7B (Touvron et al., 2023). Pruning such advanced models poses new challenges due to their semantic density, making validation on this next-generation model crucial for establishing the practical applicability of our method in rapidly evolving LLM landscapes. We compare our method with the best-performing baseline, ShortGPT. As shown in Table 4 (full results in Supplementary Table 11), our method retains 84.55% of the original performance (53.17/63.61) after pruning 9 layers, clearly outperforming ShortGPT's 62.79% (39.94/63.61) under the same compression ratio. Both results are lower than our Llama-2 7B retention (92.2%) despite the similar model size, indicating that Llama-3 is less compressible. Nevertheless, our method consistently surpasses the baseline, demonstrating its robustness across model generations.

Table 4: Comparison of pruning methods on multiple natural language benchmarks. For 8B model: Meta-Llama-3-8B-Instruct (LM), MathCoder2-Llama-3-8B (Math), Code-Llama-3-8B (Code), and Meta-Llama-3-8B (Base).

| LLM | Pruner ratio/layer | Type | Reasoning | | | Language | | | Knowledge | | | | Understanding | | | | Avg |
|---|---|---|---|---|---|---|---|---|---|---|---|---|---|---|---|---|---|
| | | | CMNLI | HeSw | PIQA | CHID | $WSC_P$ | $WSC_G$ | CSQA | BoolQ | MMLU | CMLU | $Race_H$ | $Race_M$ | XSum | C3 | |
| Llama3 -8B | Dense | Base | 32.98 | 74.67 | 80.96 | 73.78 | 56.73 | 36.54 | 73.79 | 69.97 | 64.74 | 50.79 | 63.21 | 70.54 | 3.28 | 55.18 | 57.65 |
| | | LM | 33.00 | 71.08 | 80.69 | 65.53 | 55.77 | 69.23 | 76.66 | 78.87 | 65.97 | 53.64 | 76.44 | 81.75 | 17.97 | 63.95 | 63.61 |
| | | Math | 32.99 | 71.66 | 77.97 | 57.09 | 37.50 | 58.65 | 68.22 | 69.08 | 62.08 | 45.85 | 64.75 | 69.08 | 8.68 | 53.86 | 55.53 |
| | | Code | 32.98 | 65.56 | 74.70 | 78.42 | 61.54 | 61.54 | 63.47 | 78.35 | 48.03 | 34.55 | 52.40 | 58.43 | 19.36 | 46.41 | 55.41 |
| | ShortGPT | Single | 32.83 | 45.06 | 65.78 | 23.38 | 41.35 | 53.85 | 39.56 | 63.73 | 32.37 | 28.69 | 40.14 | 45.19 | **3.68** | 43.51 | 39.94 |
| | | Merge | 32.95 | 48.58 | 64.96 | 18.43 | 36.54 | 35.58 | 42.83 | **67.22** | 33.05 | 28.71 | 30.16 | 32.45 | **3.66** | 44.27 | 37.10 |
| | Ours | | **33.42** | **54.83** | **69.75** | **34.02** | **47.12** | **62.50** | **73.79** | 64.34 | **63.13** | **50.04** | **72.81** | **77.65** | 3.00 | **46.52** | 53.78 |

## 5.4 ENHANCING LAYER-FOLDING PRUNING POTENTIAL

LaCo (Yang et al., 2024) is a merging-based pruning approach that performs within-model pruning by folding later layers into earlier ones based on activation similarity. While effective, its potential is constrained by suboptimal layer selection and merging strategies. To validate the effectiveness and potential of this type of within-model merge operation, we use our hyperparameter optimization framework with a specially designed search space (described in Section H.2). Empirically, as shown in Table 2, our framework achieves improved performance (46.26) on this configuration, unlocking greater potential from layer-folding pruning through optimized selection and merging strategies. This validates that our approach can enhance various pruning paradigms beyond cross-model scenarios, offering an effective solution when fine-tuned candidate models are unavailable.

## 5.5 COMPUTATIONAL EFFICIENCY ANALYSIS

We conducted a computational efficiency analysis against two competitive baselines, ShortGPT and LaCo, on Llama7b using post-training settings from the LLMPruner paper. We test our framework with two strategies: multi-candidate model searching (3 candidates) and single-model layer folding. We choose these strategies because they cover complementary deployment scenarios when candidate models are available versus unavailable. As shown in Supplementary Table 18, both strategies consistently outperform baselines with reduced computational overhead.

# 6  CONCLUSION

In this work, we presented a novel LLM compression approach that strategically combines layers from fine-tuned model variants instead of pruning single models. By formulating this as a zero-order optimization problem with a newly designed search space that supports layer removal, selection, and merging, our method effectively preserves model capabilities while reducing size. Experiments on Llama2-7B and Llama2-13B demonstrated that our compressed models retain 92.2% and 97.3% of original performance, respectively, despite removing $\sim 25\%$ of parameters, outperforming previous state-of-the-art methods without requiring expensive post-training. Overall, our work demonstrates that cutting and stitching layers from multiple fine-tuned variants of a model is a more effective approach to LLM compression than traditional single-model pruning. While the search complexity increases with the number of candidate models, this computational aspect represents an opportunity for future optimization techniques to further enhance efficiency.

## ACKNOWLEDGEMENTS

The authors thank Alexander Panfilov and Niccolò Ajroldi for their insightful comments and valuable feedback on this work. JG acknowledges the support of the Hector II foundation. GS acknowledges the support of the International Max Planck Research School for Intelligent Systems (IMPRS-IS). LS is supported by the NSFC Grant (No. 62576364), Shenzhen Basic Research Project (Natural Science Foundation) Basic Research Key Project (NO. JCYJ20241202124430041).

## ETHICS STATEMENT

In this work, we carefully ensure that all methods and experimental protocols conform to established ethical guidelines. Our investigation centers on layer pruning as a strategy to improve the efficiency of LLMs and to lower computational demands, contributing to more sustainable AI practices. In addition, every model and dataset employed in this research is obtained from openly accessible sources, guaranteeing respect for intellectual property and protection of personal privacy. Apart from the models used as experimental subjects (Llama2-7B, Llama-2-7B-Chat, MAmmoTH-7B, Llama-2-Coder-7B, Llama2-13B, WizardLM-13B, WizardMath-13B, Llama-2-13B-Code-Alpaca, Qwen3-8B, Qwen3-4B-Instruct, and Qwen3-4B-Thinking), we also utilized LLMs as writing assistants, as detailed in Section A. All uses of LLMs in this work comply with the ICLR Code of Ethics.

## REPRODUCIBILITY STATEMENT

We made several efforts to ensure reproducibility. First, we provide detailed experimental settings and hyperparameters used throughout this paper in Section 4, Appendix B, and Section 5.5, and report all evaluation metrics in Section 5. Second, our code will be submitted with the paper, accompanied by detailed usage instructions and scripts to reproduce all reported results.

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

## A   THE USE OF LARGE LANGUAGE MODELS

We used large language models solely as a general-purpose writing aid to help improve the clarity and readability of the text, and to suggest minor wording improvements. The LLMs did not contribute to the research ideation, experimental design, analysis, or interpretation of results. All technical content, experiments, and conclusions presented in this paper are entirely the work of the authors.

# B  BASELINE

To ensure fair comparison, we applied various baseline pruning methods including LLM-Pruner(LLMPru) (Ma et al., 2023), SliceGPT (Ashkboos et al., 2024), LaCo (Yang et al., 2024) and ShortGPT (Men et al., 2024):

**LLM-Pruner** adopts structural pruning that selectively removes non-critical coupled structures based on gradient information, maximally preserving the majority of the LLM's functionality. It applies post-training to the pruned model, for fair comparison, we do not apply post training to it.

**SliceGPT** is a post-training sparsification scheme that replaces each weight matrix with a smaller matrix, reducing the embedding dimension of the network. Specifically, they applied PCA to the hidden representation from shallow to deep layers, and incorporated the dimension reduction matrix into existing network parameters.

**LaCo** is a pruning method for large language models based on reducing layers. LaCo gradually merges similar layers from deep to shallow and sets a threshold to avoid merging too many layers.

**ShortGPT** introduced the Block Influence (BI) metric, which uses the similarity between layer's input and output to measure the importance of each layer.

# C  EVALUATION BENCHMARKS

**CMNLI (Chinese Multi-Genre Natural Language Inference) (CNLI)** consists of two parts: XNLI and MNLI. It contains text from various domains, including fiction, telephone conversations, travel, and government sources. XNLI is a cross-lingual extension of the MultiNLI corpus, professionally translated into multiple languages, including Chinese, providing a robust framework for assessing language understanding across linguistic boundaries. Models must determine whether pairs of sentences exhibit entailment, contradiction, or neutrality.

**HellaSwag (HeSw)** tests commonsense reasoning about physical situations. The dataset uses a "Goldilocks" zone of complexity where examples are obviously nonsensical to humans but challenging for state-of-the-art models. Despite being trivial for humans (>95% accuracy), even advanced models struggled with this benchmark upon its release, making it effective for measuring progress in commonsense inference.

**PIQA (Physical Interaction Question Answering)** is a multi-choice question and answer dataset that focuses on everyday scenarios, exploring models' understanding of real-world physical laws through daily situations.

**CHID (Chinese IDiom)** is an idiom cloze test focusing on the representation and selection of Chinese idioms, requiring cultural and linguistic knowledge specific to Chinese.

**WSC (Winograd Schema Challenge)** serves as a prominent benchmark for evaluating machine understanding through pronouns resolution problems that are trivial for humans but require commonsense reasoning for machines to solve correctly. The dataset consists of pairs of sentences differing in one or two words with ambiguous pronouns resolved differently in the two sentences, designed to test a system's commonsense reasoning abilities.

**CommonSenseQA (CSQA)** is a multiple-choice question answering dataset containing 12,102 questions with one correct answer and four distractor answers, requiring different types of commonsense knowledge to predict the correct answers. The dataset was constructed using ConceptNet relations and crowd-sourced questions to test commonsense reasoning.

**BoolQ** provides 15,942 yes/no questions that occur naturally in unconstrained environments, testing models' binary decision-making abilities.

**MMLU (Massive Multitask Language Understanding)** evaluates models across 57 diverse subjects covering STEM, humanities, and social sciences. The benchmark tests knowledge and problem-solving ability with content ranging from elementary to professional levels. This benchmark has become a standard evaluation metric in the field, with scores prominently reported for virtually all language models, and uses multiple-choice questions that allow for simple accuracy calculations.

**CMMLU (Chinese Massive Multitask Language Understanding) (CMLU)** Developed to address the gap in evaluating knowledge and reasoning capabilities in Chinese, CMMLU is a comprehensive benchmark covering 67 subjects from elementary to advanced professional levels across natural sciences, social sciences, engineering, and humanities. The benchmark includes topics with Chinese-specific answers that may not be universally applicable in other regions or languages, making it a fully Chinese-oriented evaluation tool.

**RACE (Reading Comprehension from Examinations)** is collected from English examinations in China designed for middle and high school students, providing a culturally diverse reading assessment.

**XSum** evaluates abstract single document summarization systems, focusing on the ability to create concise one-sentence summaries capturing the essence of articles.

**C3 (Chinese Multiple-Choice Machine Reading Comprehension)** consists of multiple-choice questions from Chinese proficiency exams and ethnic Chinese exams.

## D    TASK ARITHMETIC MERGING

Task Arithmetic Ilharco et al. (2022) enhances model capabilities through vector operations by leveraging weighted combinations of task-specific knowledge. Given a base model with weights $\theta_{\text{pre}}$ and task-specific fine-tuned weights $\{\theta_t^{\text{ft}}\}_{t=1}^n$, task vectors are defined as $\tau_t = \theta_t^{\text{ft}} - \theta_{\text{pre}}$. The merged weights are then computed through $\theta_{\text{Merge}} = \theta_{\text{pre}} + \lambda \sum_{t=1}^n \tau_t$, where $\lambda$ controls the magnitude of task-specific adaptations.

## E    DESCRIPTIONS OF SMAC-BASED MULTI-FIDELITY OPTIMIZATION

Our implementation extends SMAC (Lindauer et al., 2022), integrating Hyperband (HB) (Li et al., 2018) with Bayesian Optimization (BO) (Snoek et al., 2012) and employing Random Forest (Breiman, 2001) as the surrogate model.

The framework operates using minimum and maximum budgets ($b_{\min}$, $b_{\max}$) with a spacing parameter $\eta > 1$. The algorithm creates $s_{\max} = \lfloor \log_\eta(b_{\max}/b_{\min}) \rfloor$ brackets, each initiating with $n_i = \lfloor \eta^{s_{\max}-i} \cdot \frac{\eta}{\eta-1} \rfloor$ configurations. Within each bracket, Successive Halving proceeds through $\lfloor \log_\eta(\frac{n_i}{n_{\min}}) \rfloor + 1$ rounds, evaluating configurations at increasing budgets while progressively eliminating underperforming candidates. Specifically, after evaluating all configurations at budget $b$, only the top $\lfloor \frac{n_i}{\eta^l} \rfloor$ performers advance to the next round with an increased budget of $\eta b$.

A key enhancement is the Random Forest model that learns from all prior configuration-performance pairs, prioritizing data from higher budgets. This model guides the selection of promising configurations via Expected Improvement, balancing exploration and exploitation. As the optimization progresses, the evaluation of more configurations at higher budgets enables the algorithm to correct potential misjudgments from lower-fidelity evaluations.

For a detailed algorithmic description, see Algorithm 2, which presents the complete optimization process incorporating trial limits. This integration of multi-fidelity resource allocation with surrogate-based modeling delivers efficient configuration space exploration while maintaining evaluation quality.

## F    UNDERSTANDING STRATEGY SELECTION VIA LAYER-LEVEL ANALYSIS

To investigate how our approach works for model compression with superior performance, we analyze the architectural decisions from multiple perspectives: **the theoretical foundation of model merging**, **empirical observations of Layer-wise Patterns**, and **post-hoc analysis of layer characteristics**.

### F.0.1    THEORETICAL FOUNDATION: WHY MODEL MERGING WORKS

The underlying principle of model merging is that fine-tuned variants from a common pre-trained initialization typically converge to parameters within the same loss basin. While neural network loss functions are generally non-convex, recent work has demonstrated that parameters from different

---

**Algorithm 2** SMAC-based Multi-Fidelity Optimization

---

**Require:** Configuration space $\Theta$, minimum budget $b_{\min}$, maximum budget $b_{\max}$, spacing factor $\eta > 1$, maximum trials $T_{\max}$
**Ensure:** Optimized configuration $\theta^*$

1: $s_{\max} \leftarrow \lfloor \log_\eta(\frac{b_{\max}}{b_{\min}}) \rfloor$                 ▷ Maximum brackets
2: $\mathcal{D} \leftarrow \emptyset$                 ▷ Observation history
3: $\theta^* \leftarrow \emptyset, y^* \leftarrow \infty$             ▷ Best configuration tracking
4: $T \leftarrow 0$                 ▷ Initialize trial counter
5: **for** $i \in \{s_{\max}, s_{\max} - 1, \ldots, 0\}$ **do**
6:     **if** $T \geq T_{\max}$ **then**
7:         **break**            ▷ Exit if reached maximum trials
8:     **end if**
9:     $n_i \leftarrow \lfloor \eta^{s_{\max}-i} \cdot \frac{\eta}{\eta-1} \rfloor$           ▷ Initial configurations
10:     $\mathcal{M} \leftarrow \text{FitRandomForest}(\mathcal{D})$        ▷ Build surrogate model
11:     **if** $|\mathcal{D}| = 0$ **then**
12:         $\Theta_i \leftarrow$ Sample $n_i$ random configurations from $\Theta$
13:     **else**
14:         $\Theta_i \leftarrow$ Select $n_i$ configurations with highest EI based on $\mathcal{M}$
15:     **end if**
16:     $s_i \leftarrow \lfloor \log_\eta(\frac{n_i}{1}) \rfloor + 1$            ▷ SH rounds
17:     $\mathcal{A} \leftarrow \Theta_i$            ▷ Set of active configurations
18:     $b \leftarrow b_{\min} \cdot \eta^i$             ▷ Initial budget
19:     **for** $l \in \{0, 1, \ldots, s_i - 1\}$ **do**
20:         **if** $T \geq T_{\max}$ **then**
21:             **break**        ▷ Exit if reached maximum trials
22:         **end if**
23:         $n_{i,l} \leftarrow \lfloor \frac{n_i}{\eta^l} \rfloor$          ▷ Current pool size
24:         **for** each $\theta \in \mathcal{A}$ **do**
25:             $y_\theta \leftarrow f(\theta, b)$       ▷ Evaluate configuration
26:             $\mathcal{D} \leftarrow \mathcal{D} \cup \{(\theta, b, y_\theta)\}$      ▷ Update history
27:             $T \leftarrow T + 1$       ▷ Increment trial counter
28:             **if** $b = b_{\max}$ and $y_\theta < y^*$ **then**
29:                 $y^* \leftarrow y_\theta, \theta^* \leftarrow \theta$      ▷ Update best
30:             **end if**
31:             **if** $T \geq T_{\max}$ **then**
32:                 **break**      ▷ Exit if reached maximum trials
33:             **end if**
34:         **end for**
35:         Sort $\mathcal{A}$ by performance
36:         $\mathcal{A} \leftarrow$ Top $\lfloor \frac{n_{i,l}}{\eta} \rfloor$ configurations from $\mathcal{A}$
37:         $b \leftarrow \min(b \cdot \eta, b_{\max})$        ▷ Increase budget
38:         **if** $b = b_{\max}$ or $|\mathcal{A}| = 1$ **then**
39:             **break**
40:         **end if**
41:     **end for**
42: **end for**
43: **return** $\theta^*$

---

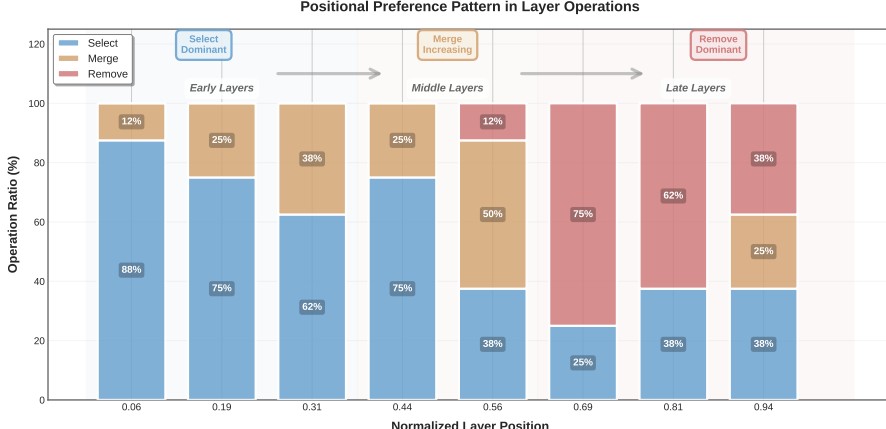

Figure 4: Positional preference pattern in layer operations. Operation distribution across normalized layer positions, averaged over 7B and 13B models. Early layers favor SELECT, middle layers favor MERGE, and late layers favor REMOVE.

training runs can be interpolated without increasing loss, a phenomenon known as mode connectivity Garipov et al. (2018); Frankle et al. (2020).

Garipov et al. (2018) showed that different optima can be connected by simple curves with nearly constant accuracy. Frankle et al. (2020) further demonstrated that networks sharing part of their optimization trajectory converge to linearly connected regions, where the linear interpolation $\theta(t) = (1 - t)\theta_A + t\theta_B$ maintains low loss for all $t \in [0, 1]$. Entezari et al. (2021) conjectured that when accounting for permutation invariance, SGD solutions exhibit no barrier in linear interpolation.

Crucially, fine-tuned models initialized from the same pre-trained model $\theta_0$ share a significant portion of their optimization trajectory, enabling merging without explicit permutation alignment Wortsman et al. (2022); Ilharco et al. (2022). This is evidenced by the small Frobenius distance between such models Yadav et al. (2023):

$$\|\theta_t^{(i)} - \theta_t^{(j)}\|_F \ll \|\theta_t^{(i)} - \theta_0\|_F \tag{3}$$

where $\theta_t^{(i)}$ and $\theta_t^{(j)}$ denote models fine-tuned on different tasks $i$ and $j$. This proximity in parameter space, combined with the wide, flat minima characteristic of fine-tuned models, provides the theoretical foundation for merging complementary capabilities while maintaining performance.

These properties make merging a natural guide for pruning because the shared loss basin reveals redundant or overlapping layers whose removal does not disrupt the model's performance.

### F.0.2 EMPIRICAL OBSERVATIONS: LAYER-WISE PATTERNS

**Pattern 1: Positional preference.** To identify systematic patterns in layer operations, we normalize layer positions (position = layer_index / total_layers) and partition the space [0,1] into eight bins, computing operation percentages averaged across 7B and 13B families. Figure 4 reveals a clear pattern: early layers favor SELECT (64.1% at position 0-0.3), middle layers favor MERGE (49.9% at 0.3-0.6), and late layers favor REMOVE (70.7% at 0.6-1.0).

**Pattern 2: Robustness (redundancy) scales with model size**. From the visualization in Fig.2, we can see that the 13B model shows a simpler structure, which is mainly merged with LM models, while the 7B model shows a more complex structure utilizing mixed and specialized models. This suggests that as model size decreases, more diverse mixing strategies may be needed to maintain performance. This architectural difference, coupled with the superior preservation rate of the 13B model compared to the 7B model, demonstrates that robustness (redundancy) scales with model size.

### F.0.3 WHY DO THESE PATTERNS EMERGE?

We now investigate whether these decisions align with interpretable layer properties. We conduct post-hoc analysis examining two complementary perspectives: cross-model representational alignment and within-model layer redundancy. All analyses below are conducted using Llama2-7B models.

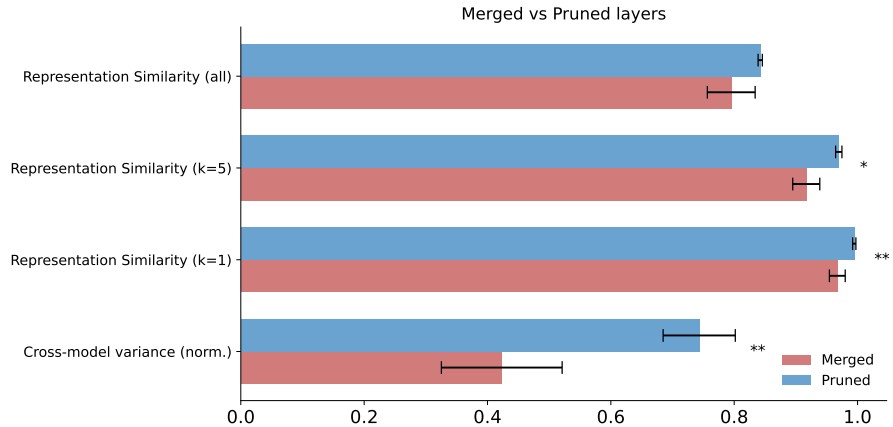

Figure 5: Comparison of CKA (Centered Kernel Alignment)-based features between merged and pruned layers with the mean and standard error. Stars indicate statistical significance (*: $p < 0.05$, **: $p < 0.01$). Although both merged and pruned layers both exhibit high similarity with their neighboring layers, merged layers maintain lower cross-model variance and stronger local and global CKA coherence, while pruned layers exhibit higher representational divergence.

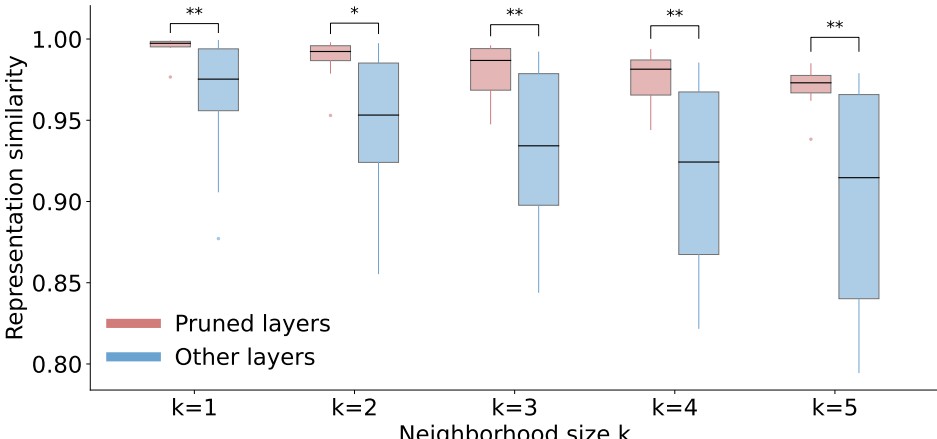

Figure 6: Representation similarity of pruned versus retained layers across neighborhood sizes (k=1) to (k=5). Pruned layers (red) exhibit significantly higher similarity than retained layers (blue), indicating that pruning primarily removes redundant layers (**$p < 0.01$).

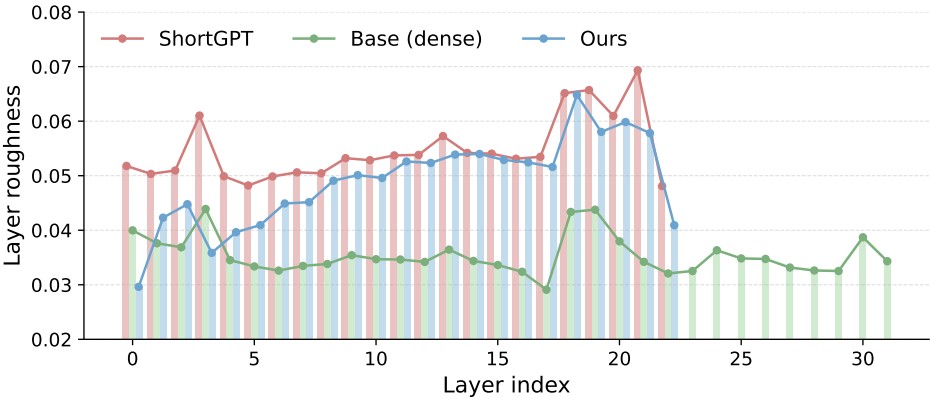

Figure 7: Layer-wise roughness comparison. Our method produces consistently lower roughness scores than ShortGPT, indicating smoother and more coherent depth-wise representation transitions.

We extracted CKA (Centered Kernel Alignment)-based features Kornblith et al. (2019) that capture both inter-model and intra-model structural properties. First, *cross-model variance* quantifies representational divergence across the four models by measuring the variability of their pairwise CKA similarities at each layer; higher variance indicates that models have learned different representations at that depth. Second, we compute *local CKA* at multiple scales (k = 1, 5, and all layers), measuring how consistently each layer aligns with its immediate neighbors (surrounding block). These metrics capture short-range smoothness, block-level coherence, and global structural integration. Lower local CKA values indicate better representational consistency at the corresponding scale.

**Analysis 1: Merged layers exhibit cross-model alignment.** Across all extracted features, merged layers consistently show higher representational consistency both within and across models (Fig. 5). They exhibit lower cross-model variance, indicating that all four models converge to similar feature representations at these depths. Their lower local CKA values further suggest that the representations of merged layers can be aligned and fused without structural conflict. In contrast, pruned layers demonstrate higher cross-model variance and lower global consistency, revealing that different models encode incompatible representations in these regions, retaining them during fusion contributes little useful information and may introduce conflicts.

**Analysis 2: Pruned layers show within-model redundancy.** We examine within-model layer similarity by measuring how closely each layer's representations match its neighbors at different scales (k=1 to k=5). Results in Fig. 6 demonstrate that pruned layers (red) consistently exhibit higher representation similarity to their neighbors compared to retained layers (blue) across all neighborhood sizes. Statistical significance tests (**) confirm that the representations of redundant layers are highly similar to adjacent layers and thus contribute minimal unique information.

**Analysis 3: Our method maintains smoother transitions than baselines.** We compared the representation similarity of the pruned model produced by ShortGPT with that of our searched model. For each model, we computed the CKA similarity around each layer change relative to its neighboring layers. As shown in Fig. 7, the merge-based model consistently yields lower roughness values, indicating smoother and more coherent depth-wise representation transitions. This suggests that our method preserves the natural progression of representations instead of disrupting the hierarchical flow. In contrast, ShortGPT introduces sharper local changes, leading to a more fragmented representational structure. The smoother similarity profile of our model demonstrates a more stable internal organization, with fewer disruptive shifts between layers.

Together, these analyses reveal that merging and pruning target fundamentally different structural properties. Merging capitalizes on *cross-model consensus*: layers where all models have converged to similar representations can be safely fused. Pruning exploits *within-model redundancy*: layers that duplicate information already present in their neighbors can be removed without information loss. Critically, these patterns are *discovered* rather than *designed*. Our optimization framework identifies them automatically by maximizing performance under compression constraints. The strong correlation between discovered patterns and interpretable layer properties validates that our method captures genuine structural regularities rather than exploiting dataset-specific artifacts.

## G    FULL BASELINE RESULTS

To validate the efficiency of our proposed method, we conducted comparative experiments against established baseline techniques. For fair comparison with other baseline methods, we selected the same pruning ratios matching those used in LaCo (Yang et al., 2024) and ShortGPT (Men et al., 2024) while being lower than those of other approaches. In order to make a fairer comparison, we reproduced all the results and evaluated them on OpenCompass (Contributors, 2023) as in LaCo.All experiments run on NVIDIA Tesla A100 GPUs. For each baseline method, we explored three scenarios: (1) applying each baseline pruning method individually to all candidate models, (2) first pruning each candidate model using existing methods and then merging them, and (3) first merging the candidate models and then applying pruning techniques.

We use the official implement of LLM-pruner and LaCo, It's worth noting that when reproducing the LaCo method, we referenced the hyperparameter settings from the original paper. Due to differences in hardware, we couldn't fully reproduce the paper's results: we couldn't obtain models with pruning ratios consistent with the paper using the provided hyperparameters. We maintained consistency in

all other parameters while gradually adjusting the threshold from 0.75 until achieving the desired pruning ratio. The specific parameters are detailed in the Table 5.

For the reproduction of ShortGPT, we implemented the algorithm based on the original paper and similarly sampled 10,000 instances from the PG19 (Rae et al., 2019) dataset as calibration data, following the methodology described in the paper. The resulting removed layers are shown in the Table. The removed layers for the base model align with those reported in the ShortGPT paper, albeit in a different sequence. We attribute this variation to slight differences in calculated layer importance scores. The specific configuration of removed layers for each model is detailed in the Table 6.

For the merging process, we employed task arithmetic with weighting parameters in the range of [0.5, 1.0]. The full results of the baseline methods on the 7B model and the 13B model are presented in Table 7 and Table 8, respectively.

Table 5: Hyperparameter settings for LaCo results. $\mathcal{C}$: Number of layers combined in each merge; $\mathcal{L},\mathcal{H}$: Layer range $[\mathcal{L}, \mathcal{H}]$; $\mathcal{I}$: Minimum interval between two adjacent merged layers; $\mathcal{T}$: Threshold for representation similarity.

| Size | Model | $\mathcal{C}$ $\mathcal{L}$ $\mathcal{H}$ $\mathcal{I}$ $\mathcal{T}$ |
|---|---|---|
| Llama2-13B | Llama-2-13B | 6 1 40 2 0.7 |
| | WizardLM-13B | 6 1 40 2 0.65 |
| | WizardMath-13B | 6 1 40 2 0.7 |
| | llama-2-13b-code-alpaca | 6 1 40 2 0.7 |
| | Merge-then-prune | 6 1 40 2 0.65 |
| | Prune-then-merge | 6 1 40 2 0.65 |
| Llama2-7B | Llama-2-7B | 6 1 40 2 0.7 |
| | Llama-2-7B-Chat | 6 1 40 2 0.65 |
| | MAmmoTH-7B | 6 1 40 2 0.7 |
| | Llama-2-Coder-7B | 6 1 40 2 0.7 |
| | Merge-then-prune | 6 1 40 2 0.65 |
| | Prune-then-merge | 6 1 40 2 0.65 |

Table 6: Setup of Removed Layers for Candidate Models in ShortGPT.

| Model | Removed Layers |
|---|---|
| Llama-2-7B | 25, 27, 24, 28, 26, 29, 23, 22, 21 |
| Llama-2-7B-Chat | 27, 25, 24, 28, 29, 26, 23, 22, 21 |
| MAmmoTH-7B | 27, 25, 24, 28, 29, 23, 26, 22, 21 |
| Llama-2-Coder-7B | 27, 25, 24, 28, 29, 26, 23, 21, 22 |
| Llama-2-13B | 33, 32, 31, 30, 34, 35, 29, 28, 27, 26 |
| WizardLM-13B | 33, 32, 31, 30, 34, 35, 29, 28, 27, 36 |
| WizardMath-13B | 33, 31, 32, 30, 34, 35, 29, 28, 27, 36 |
| llama-2-13b-code-alpaca | 33, 31, 32, 30, 34, 35, 29, 28, 27, 26 |

# H   ADDITIONAL ANALYSIS

## H.1   DIFFERENT CALIBRATION DATASETS AND METRICS

In this study, we leverage multiple-choice datasets as calibration data and optimize for accuracy in a multi-objective setting. In this section, we further analyze the impact of these design choices by comparing single-objective optimization and PPL-based optimization:

Table 7: The main results of baseline methods on the 7B model across multiple natural language benchmarks using candidate models: Llama-2-7B-Chat (LM), MAmmoTH-7B (Math), Llama-2-Coder-7B (Code), and Llama-2-7B (base). "PTM" (Pruning-then-Merging) refers to first pruning each candidate model using current pruner and then merging them. "MTP" (Merging-then-Pruning) refers to first merging the candidate models and then applying pruning. For LLMPruner and SliceGPT, alignment challenges exist after pruning. LLMPruner removes different model blocks, while SliceGPT calculates orthogonal transformation matrices that are highly dependent on each model's specific weight distributions and activation patterns, resulting in incompatible transformation spaces. Therefore, we only implemented "merge then prune".

| LLM | Pruner (ratio/layer) | Type | Reasoning | | | Language | | | Knowledge | | | | Understanding | | | | Avg |
|---|---|---|---|---|---|---|---|---|---|---|---|---|---|---|---|---|---|
| | | | CMNLI | HeSw | PIQA | CHID | WSC$_P$ | WSC$_G$ | CSQA | BoolQ | MMLU | CMLU | Race$_H$ | Race$_M$ | XSum | C3 | |
| Llama-7B | | Dense | Base | | | | | | | | | | | | | | | |
| | | | Base | 32.98 | 71.34 | 78.18 | 41.56 | 37.50 | 38.46 | 55.04 | 70.70 | 46.67 | 31.88 | 35.53 | 33.36 | 19.55 | 43.84 | 45.47 |
| | | | Math | 32.99 | 68.60 | 75.79 | 39.71 | 39.42 | 36.54 | 50.78 | 69.36 | 43.04 | 32.16 | 30.36 | 36.42 | 20.88 | 43.45 | 44.25 |
| | | | LM | 31.30 | 71.28 | 75.95 | 36.11 | 63.46 | 59.62 | 64.29 | 74.77 | 48.30 | 33.93 | 52.52 | 55.22 | 22.45 | 47.56 | 52.63 |
| | | | Code | 32.99 | 70.27 | 78.62 | 41.61 | 36.54 | 41.35 | 57.41 | 71.04 | 46.22 | 32.20 | 41.25 | 39.69 | 18.79 | 46.25 | 46.73 |
| | LLMPruner (25.32%) | Base | 33.00 | 58.72 | 72.25 | 29.52 | 41.35 | 0.00 | 19.74 | 57.25 | 23.69 | 25.49 | 22.07 | 21.10 | 14.67 | 28.11 | 31.93 |
| | | LM | 34.94 | 59.25 | 72.85 | 22.28 | 43.27 | 9.62 | 19.41 | 57.61 | 23.77 | 24.51 | 21.78 | 22.42 | 16.32 | 28.66 | 32.62 |
| | | MATH | 32.99 | 55.74 | 70.84 | 25.82 | 37.50 | 21.15 | 18.84 | 54.31 | 24.77 | 25.20 | 22.87 | 23.89 | 10.91 | 28.00 | 32.35 |
| | | Code | 32.99 | 59.57 | 73.34 | 30.32 | 46.15 | 0.00 | 20.15 | 57.28 | 23.21 | 25.16 | 21.56 | 21.52 | 15.19 | 31.07 | 32.68 |
| | | MTP | 34.71 | 60.57 | 73.50 | 26.62 | 40.38 | 5.77 | 19.90 | 52.14 | 24.01 | 25.30 | 23.07 | 22.98 | 15.51 | 32.49 | 32.64 |
| | SliceGPT (26.33%) | Base | 31.08 | 42.90 | 61.43 | 19.53 | 36.54 | 0.00 | 20.88 | 37.95 | 24.78 | 24.78 | 21.24 | 21.73 | 6.58 | 37.42 | 27.63 |
| | | LM | 31.70 | 43.50 | 61.37 | 18.28 | 40.38 | 0.96 | 21.21 | 38.96 | 25.56 | 25.28 | 21.93 | 22.42 | 13.13 | 38.36 | 28.79 |
| | | MATH | 31.89 | 41.55 | 58.81 | 18.43 | 39.42 | 4.81 | 19.49 | 40.09 | 25.38 | 25.02 | 25.59 | 26.88 | 8.78 | 39.56 | 28.98 |
| | | Code | 31.81 | 44.02 | 63.17 | 18.48 | 36.54 | 13.46 | 19.74 | 37.92 | 24.71 | 25.22 | 21.41 | 21.66 | 2.59 | 38.19 | 28.49 |
| | | MTP | 32.85 | 37.61 | 57.56 | 17.33 | 53.85 | 2.88 | 19.41 | 42.66 | 25.22 | 24.68 | 25.21 | 24.72 | 12.78 | 40.22 | 29.78 |
| | LACO | Base | 32.85 | 53.33 | 68.23 | 31.62 | 36.54 | 4.81 | 20.39 | 62.02 | 26.60 | 25.27 | 24.70 | 23.61 | 9.38 | 42.47 | 32.99 |
| | | LM | 32.97 | 55.24 | 69.53 | 31.47 | 36.54 | 34.62 | 22.11 | 67.22 | 29.08 | 26.16 | 28.53 | 28.27 | 14.68 | 43.51 | 37.14 |
| | | Math | 32.97 | 55.24 | 69.53 | 31.47 | 50.00 | 34.62 | 22.11 | 67.22 | 29.44 | 26.16 | 22.53 | 23.68 | 14.68 | 39.34 | 37.07 |
| | | Code | 32.28 | 53.68 | 69.15 | 32.22 | 36.54 | 1.92 | 20.56 | 61.99 | 26.31 | 25.43 | 27.10 | 22.70 | 11.14 | 43.07 | 33.15 |
| | | MTP | 32.43 | 57.80 | 71.82 | 28.97 | 41.35 | 16.35 | 27.52 | 71.28 | 30.49 | 26.88 | 25.76 | 27.09 | 8.27 | 44.33 | 36.45 |
| | | PTM | 31.89 | 56.26 | 71.22 | 27.32 | 39.42 | 22.12 | 23.42 | 72.66 | 29.30 | 26.00 | 25.19 | 26.81 | 16.11 | 43.62 | 36.52 |
| | ShortGPT (27.1%) | Base | 33.09 | 57.42 | 66.54 | 21.53 | 56.73 | 48.08 | 52.5 | 67.34 | 43.68 | 28.31 | 32.53 | 31.69 | 12.40 | 39.45 | 42.24 |
| | | LM | 33.85 | 53.93 | 63.82 | 14.59 | 39.42 | 22.12 | 58.48 | 67.95 | 35.85 | 26.60 | 48.03 | 51.18 | 6.93 | 37.21 | 40.00 |
| | | MATH | 33.97 | 56.69 | 63.38 | 17.78 | 54.81 | 44.23 | 37.26 | 69.82 | 30.68 | 25.26 | 28.24 | 30.29 | 8.26 | 31.67 | 38.02 |
| | | Code | 32.74 | 56.69 | 65.07 | 17.78 | 58.65 | 35.58 | 53.24 | 67.52 | 44.82 | 28.92 | 35.62 | 37.53 | 14.32 | 40.66 | 42.08 |
| | | MTP | 34.10 | 54.18 | 64.42 | 16.83 | 61.54 | 36.54 | 55.61 | 73.21 | 36.84 | 25.61 | 42.94 | 45.89 | 10.12 | 35.73 | 42.40 |
| | | PTM | 34.10 | 54.18 | 64.42 | 16.83 | 61.54 | 36.54 | 55.61 | 73.21 | 36.84 | 25.61 | 42.94 | 45.89 | 10.12 | 35.73 | 42.40 |

**Single Objective (Single-obj).** We used the MMLU validation dataset for calibration and kept accuracy as the optimization objective. We evaluated the resulting pruned models across our benchmark suite. As shown in Table 14, although these models still remain competitive (45.62 average), the single-objective optimization led to a noticeable decline from our multi-objective approach (48.55 vs.45.62). Importantly, the single-objective models demonstrated stronger performance on MMLU-related tasks but showed performance degradation on certain other tasks due to their narrow optimization focus. This confirms our hypothesis that broad, multi-objective optimization is necessary to preserve the broad functionality of modern LLMs, rather than overfitting to a single task domain.

**Perplexity Objective (PPL-obj).** We additionally evaluate with perplexity (PPL) on WikiText (Merity et al., 2016) as a search metric, using 1500 examples for calibration. As shown in Table 14, the resulting pruned models achieve only 25.38 on average, revealing a substantial performance drop relative to all other configurations. Even when compared to the single-objective MMLU optimization (which uses a similarly sized dataset), the PPL-optimized models showed considerably weaker performance across most tasks. These results show that, although perplexity is a common metric for language model evaluation, it is not an effective signal for preserving model capabilities during pruning, especially for tasks that require reasoning or knowledge application beyond fluent text generation.

## H.2 ENHANCING LAYER-FOLDING PRUNING POTENTIAL

We design a search space for Layer-Folding Pruning consisting of: (1) A binary selection vector $\mathbf{s} = [s_1, s_2, \ldots, s_k]$ indicating which layers to remove, and (2) An importance weight vector $\mathbf{w} =$

Table 8: The main results of baseline methods on the 13B model across multiple natural language benchmarks using candidate models: WizardLM-13B (LM), WizardMath-13B (Math), llama-2-13b-code-alpaca (Code), and Llama-2-13B (Base). "PTM" (Pruning-then-Merging) refers to first pruning each candidate model using the current pruner and then merging them. "MTP" (Merging-then-Pruning) refers to first merging the candidate models and then applying pruning. For LLMPruner and SliceGPT, alignment challenges exist after pruning. LLMPruner removes different model blocks, while SliceGPT calculates orthogonal transformation matrices that are highly dependent on each model's specific weight distributions and activation patterns, resulting in incompatible transformation spaces. Therefore, we only implemented "merge then prune"

| LLM | Pruner ratio/layer | Type | Reasoning | | | Language | | | Knowledge | | | | Understanding | | | | Avg |
|---|---|---|---|---|---|---|---|---|---|---|---|---|---|---|---|---|---|
| | | | CMNLI | HeSw | PIQA | CHID | $WSC_P$ | $WSC_G$ | CSQA | BoolQ | MMLU | CMLU | $Race_H$ | $Race_M$ | XSum | C3 | |
| Llama -13B | | Base | 32.99 | 74.77 | 79.71 | 47.35 | 50.96 | 63.46 | 67.24 | 71.38 | 55.84 | 38.74 | 57.98 | 60.17 | 23.47 | 47.51 | 55.11 |
| | Dense | LM | 35.36 | 70.41 | 78.73 | 36.21 | 57.69 | 60.58 | 65.03 | 73.70 | 53.48 | 30.85 | 66.12 | 71.66 | 22.44 | 52.00 | 55.30 |
| | | MATH | 32.99 | 68.78 | 77.26 | 44.36 | 36.54 | 19.23 | 60.36 | 78.44 | 54.21 | 38.12 | 47.74 | 48.82 | 19.51 | 44.66 | 47.93 |
| | | Code | 32.99 | 74.82 | 80.14 | 47.30 | 51.92 | 63.46 | 68.88 | 72.72 | 55.92 | 39.26 | 58.03 | 63.72 | 24.45 | 48.38 | 55.86 |
| | | Base | 33.27 | 63.57 | 75.41 | 34.17 | 37.50 | 0.00 | 19.57 | 45.35 | 23.08 | 25.36 | 21.61 | 21.80 | 14.41 | 29.64 | 31.77 |
| | LLMPruner (21.2%) | LM | 33.49 | 60.28 | 75.57 | 23.68 | 39.42 | 0.00 | 19.00 | 63.24 | 23.27 | 25.23 | 22.36 | 21.45 | 17.13 | 32.00 | 32.58 |
| | | MATH | 32.99 | 55.49 | 72.91 | 30.02 | 41.35 | 0.00 | 19.08 | 53.18 | 23.06 | 25.53 | 21.36 | 21.31 | 12.25 | 29.10 | 31.26 |
| | | Code | 33.18 | 64.21 | 75.52 | 34.17 | 43.27 | 0.00 | 19.90 | 47.80 | 23.19 | 25.52 | 21.61 | 22.08 | 16.08 | 29.59 | 32.58 |
| | | MTP | 33.86 | 64.11 | 73.50 | 22.18 | 60.58 | 0.00 | 21.46 | 61.96 | 23.84 | 25.62 | 22.16 | 21.59 | 14.98 | 32.11 | 34.14 |
| | | Base | 30.39 | 46.69 | 63.22 | 18.78 | 42.31 | 25.96 | 25.23 | 37.83 | 30.43 | 25.14 | 23.47 | 24.65 | 8.78 | 39.56 | 31.60 |
| | SliceGPT (23.6%) | LM | 33.19 | 42.44 | 59.90 | 18.03 | 54.81 | 19.23 | 32.51 | 41.22 | 33.09 | 25.75 | 29.45 | 29.87 | 9.99 | 37.75 | 33.37 |
| | | MATH | 32.73 | 36.27 | 59.30 | 17.38 | 42.31 | 0.00 | 21.62 | 37.83 | 30.33 | 25.16 | 23.84 | 24.16 | 1.54 | 40.82 | 28.09 |
| | | Code | 30.82 | 46.69 | 63.00 | 19.18 | 42.31 | 27.88 | 24.82 | 37.83 | 31.38 | 25.20 | 23.47 | 24.65 | 8.83 | 40.00 | 31.86 |
| | | MTP | 30.98 | 46.83 | 62.57 | 19.33 | 51.92 | 49.04 | 37.76 | 38.38 | 33.55 | 25.22 | 23.53 | 23.05 | 9.95 | 39.67 | 35.13 |
| | | Base | 32.97 | 59.38 | 73.45 | 36.26 | 37.50 | 37.50 | 19.41 | 57.31 | 25.03 | 24.41 | 22.47 | 23.19 | 16.39 | 37.92 | 35.94 |
| | LaCo (24.6%) | LM | 32.33 | 60.18 | 70.57 | 32.67 | 34.62 | 34.62 | 52.58 | 62.66 | 36.26 | 25.80 | 60.38 | 62.53 | 8.79 | 49.21 | 44.51 |
| | | Math | 33.97 | 56.51 | 72.25 | 33.52 | 44.23 | 44.23 | 21.38 | 64.19 | 25.35 | 24.55 | 21.98 | 21.94 | 12.77 | 37.48 | 36.74 |
| | | Code | 32.99 | 59.53 | 75.03 | 38.41 | 51.92 | 0.00 | 19.49 | 53.18 | 24.48 | 24.72 | 22.87 | 22.28 | 17.70 | 37.53 | 34.30 |
| | | MTP | 33.49 | 62.50 | 74.37 | 35.26 | 63.46 | 63.46 | 18.84 | 64.65 | 41.83 | 24.87 | 26.10 | 25.97 | 15.93 | 39.51 | 42.16 |
| | | PTM | 31.85 | 29.80 | 51.31 | 12.74 | 36.54 | 36.54 | 19.57 | 62.08 | 24.37 | 25.19 | 22.10 | 22.77 | 0.40 | 35.12 | 29.31 |
| | | Base | 32.99 | 67.07 | 73.45 | 36.46 | 42.31 | 45.19 | 66.99 | 58.56 | 54.74 | 38.39 | 56.89 | 54.06 | 18.58 | 46.19 | 49.42 |
| | ShortGPT (24.6%) | LM | 32.95 | 62.64 | 73.50 | 28.22 | 36.54 | 50.96 | 65.44 | 67.71 | 53.50 | 30.73 | 65.52 | 71.38 | 19.12 | 48.60 | 50.49 |
| | | MATH | 32.99 | 59.63 | 70.40 | 31.12 | 40.38 | 1.92 | 59.71 | 70.00 | 52.70 | 36.94 | 43.51 | 44.29 | 7.73 | 43.84 | 42.51 |
| | | Code | 32.92 | 67.03 | 74.37 | 36.41 | 55.77 | 46.15 | 68.96 | 60.55 | 54.94 | 38.30 | 53.60 | 58.57 | 8.41 | 47.18 | 50.23 |
| | | MTP | 31.07 | 63.24 | 68.61 | 27.17 | 49.04 | 43.27 | 65.68 | 78.01 | 51.26 | 36.88 | 57.38 | 62.67 | 16.94 | 44.05 | 49.66 |
| | | PTM | 31.08 | 63.32 | 68.66 | 27.12 | 49.04 | 43.27 | 65.68 | 77.98 | 51.23 | 36.82 | 57.40 | 62.47 | 17.01 | 43.95 | 49.65 |

$[w_1, w_2, \ldots, w_k]$ representing each layer's importance value. Retained layer $L_i'$ performs a depth-wise linear combination with itself and adjacent removed layers:

$$L_i' = \beta_i \cdot L_i + \sum_{j \in \mathcal{N}(i)} \beta_j \cdot L_j \cdot \mathbb{1}_{s_j=1}$$

where $\mathcal{N}(i)$ represents adjacent layers to $L_i$, $\mathbb{1}_{s_j=1}$ indicates layer $j$ is removed, and $\beta_j$ are normalized weights derived from $\mathbf{w}$ such that $\beta_i + \sum_{j \in \mathcal{N}(i)} \beta_j \cdot \mathbb{1}(s_j = 1) = 1$. This ensures retained layers incorporate information from nearby removed layers, preserving network functionality.

## H.3 EFFICIENCY ANALYSIS

**Budget allocation to search trials.** Our optimizer dynamically adjusts the budget allocation during the search process, where the budget is defined as the calibration dataset size used for search. As the allocation of search trials directly determines the overall search duration. Here, we analyze the budget distribution during the search process, as shown in Table 15. Our analysis reveals that only 22% of the search trials utilize the full budget, while over 41.4% of the evaluations were conducted with the minimum budget, which is 5-10 times smaller. This efficient allocation enables our pruning to significantly increase the chance of discovering superior configurations under the same computational budget.

Table 9: Performance comparison of various model pruning strategies across multiple benchmark categories. The settings include LR-only (Layer Removal only), LS+LR (combined Layer Selection and Layer Removal), FL-merge (Folding Layers Merging), Single-obj (Single-objective optimization), and PPL-obj (Perplexity-based objective). For multi-objective optimization approaches, three representative Pareto-optimal solutions (numbered 1-3) are showed.

| setting | Reasoning | | | Language | | | Knowledge | | | | Understanding | | | | Avg |
|---|---|---|---|---|---|---|---|---|---|---|---|---|---|---|---|
| | CNLI | HeSw | PIQA | CHID | WSC$_P$ | WSC$_G$ | CSQA | BoolQ | MMLU | CMLU | Race$_H$ | Race$_M$ | XSum | C3 | |
| LR-only-LM-1 | 33.93 | 57.51 | 65.49 | 18.18 | 62.46 | 48.03 | 58.79 | 62.18 | 45.76 | 30.95 | 49.54 | 53.36 | 1.45 | 38.60 | 44.73 |
| LR-only-LM-2 | 33.58 | 52.10 | 64.25 | 19.53 | 50.00 | 62.50 | 63.64 | 41.80 | 48.33 | 32.84 | 51.03 | 51.46 | 5.47 | 39.56 | 44.01 |
| LR-only-LM-3 | 34.96 | 53.80 | 66.70 | 18.58 | 49.04 | 58.65 | 60.61 | 68.87 | 47.85 | 33.54 | 42.51 | 43.04 | 8.05 | 41.42 | 44.83 |
| LR-only-Math-1 | 33.77 | 54.49 | 68.23 | 21.93 | 62.50 | 37.50 | 27.85 | 57.52 | 37.08 | 28.73 | 31.42 | 34.05 | 7.51 | 37.92 | 38.61 |
| LR-only-Math-2 | 31.69 | 56.56 | 68.77 | 27.07 | 63.46 | 30.77 | 36.69 | 62.35 | 39.17 | 29.15 | 33.39 | 38.65 | 4.41 | 43.34 | 40.39 |
| LR-only-Math-3 | 32.94 | 58.43 | 69.64 | 25.97 | 54.81 | 25.96 | 29.89 | 62.84 | 33.46 | 26.92 | 31.39 | 32.10 | 8.06 | 40.16 | 38.04 |
| LR-only-Code-1 | 30.13 | 57.60 | 70.35 | 27.07 | 63.46 | 11.54 | 50.94 | 65.96 | 42.64 | 30.96 | 36.39 | 36.77 | 3.15 | 43.78 | 40.77 |
| LR-only-Code-2 | 34.94 | 57.37 | 68.55 | 28.67 | 42.31 | 41.35 | 54.46 | 63.00 | 42.49 | 27.39 | 34.88 | 35.31 | 4.08 | 43.78 | 41.33 |
| LR-only-Code-3 | 34.93 | 56.71 | 69.42 | 25.92 | 59.62 | 31.65 | 52.83 | 62.20 | 43.03 | 28.80 | 38.51 | 39.07 | 2.87 | 41.70 | 41.95 |
| LR-only-Base-1 | 32.67 | 54.21 | 66.00 | 26.07 | 36.54 | 1.92 | 49.47 | 64.19 | 44.47 | 28.84 | 38.99 | 38.86 | 0.25 | 41.59 | 37.43 |
| LR-only-Base-2 | 32.22 | 56.48 | 67.46 | 26.32 | 61.54 | 50.00 | 41.44 | 66.91 | 40.54 | 28.01 | 37.94 | 39.35 | 0.96 | 41.92 | 42.22 |
| LR-only-Base-3 | 31.13 | 52.90 | 67.95 | 27.97 | 36.54 | 0.00 | 54.63 | 64.13 | 43.01 | 30.03 | 35.56 | 37.05 | 6.79 | 41.70 | 37.81 |
| FL-merge-1 | 32.99 | 52.90 | 63.66 | 19.28 | 46.15 | 62.50 | 60.52 | 75.20 | 48.30 | 34.33 | 50.77 | 55.29 | 6.39 | 39.40 | 46.26 |
| FL-merge-2 | 32.99 | 51.99 | 63.44 | 18.33 | 46.15 | 63.46 | 61.26 | 74.77 | 48.80 | 33.84 | 51.11 | 56.34 | 5.75 | 37.86 | 46.15 |
| FL-merge-3 | 33.89 | 51.15 | 62.62 | 18.63 | 50.00 | 61.54 | 60.44 | 75.78 | 48.61 | 33.96 | 50.74 | 55.85 | 5.72 | 38.03 | 46.15 |
| LS+LR-1 | 34.75 | 53.65 | 66.32 | 17.83 | 63.46 | 22.12 | 59.71 | 70.61 | 47.32 | 33.77 | 36.62 | 33.91 | 8.54 | 42.35 | 42.21 |
| LS+LR-2 | 31.74 | 55.25 | 68.39 | 26.77 | 63.46 | 10.58 | 58.72 | 66.27 | 47.40 | 33.15 | 40.02 | 45.26 | 2.62 | 44.16 | 42.41 |
| LS+LR-3 | 32.92 | 55.84 | 65.07 | 17.98 | 63.46 | 26.92 | 58.97 | 51.22 | 48.97 | 34.61 | 48.68 | 49.44 | 8.33 | 42.41 | 43.20 |
| Single-obj | 32.15 | 56.02 | 67.46 | 19.08 | 39.42 | 48.08 | 62.33 | 74.43 | 47.40 | 34.14 | 50.94 | 52.86 | 12.35 | 41.97 | 45.62 |
| PPL-obj | 33.39 | 23.89 | 52.07 | 14.84 | 45.19 | 7.69 | 19.33 | 39.51 | 24.25 | 24.69 | 22.81 | 21.17 | 0.06 | 26.36 | 25.38 |

**Wall-Clock Time Analysis of the Search Process.** There are three main phases of our search process to consider for computational costs. **1) Computation of a new merge:** This phase involves computing a new candidate point to evaluate later with the search procedure. For standard merging algorithms, such as task arithmetic (which we use in the submission), the cost of the merge is negligible, coming down to approximately 3 operations per model parameter. The operation can run on the accelerator (e.g., GPU) when memory permits, or be executed with minimal CPU RAM by streaming parameter blocks from disk. Although it can be overlapped with the next step, it was sufficiently fast in practice (e.g., merging two 7B models on GPU takes only 11.2 seconds), we did not implement this overlap, and there remains room for further optimization. **2) Evaluation of the merge:** Next, the merged point is evaluated, i.e., we measure the accuracy of this checkpoint on our training task. The cost of this operation is a function of (a) the size of the evaluation set and (b) the type of evaluation, both of which influence the speed. However, as this step is not specific to our approach, any inference framework for fast evaluation, such as vLLM, can be used (as we do). For example, evaluating PIQA requires only prefilling. With vLLM on our GPU V100 (batch size = 16), it takes us 21.23 seconds to evaluate on 1000 samples. We also note that, due to our multi-fidelity search approach, we can often end the evaluation early and do not need to check the full dataset (see Table 15). **3) Updating the coefficients of Bayesian hyperparameter estimation:** We use SMAC, a well-established and optimized package for Bayesian hyperparameter optimization. As the estimation is based on random forests, it is very cheap to update. For us, one step of the update takes 2.6 seconds.

With our parallel acceleration strategies (e.g., simultaneous merging and evaluation), the evaluation phase becomes the dominant factor in end-to-end runtime. To provide a clearer picture, we report the evaluation wall-clock time for different datasets under various computational budgets on Llama2-7B as shown in Table 16. Model initialization using vLLM takes 19.52 seconds. Overall, our approach takes 30/35.36/60.36 seconds per round across different fidelity levels, and we run 500 rounds in total, with 41% of trials requiring only the smallest budget. When evaluation parallelism is disabled (parallelism = 1), the total wall-time is simply the sum of the individual evaluation times.

Table 10: Model Performance Comparison Across Pruning Ratios

| Model | Prune Ratio | Reasoning | | | Language | | | Knowledge | | | | Understanding | | | | Avg |
|---|---|---|---|---|---|---|---|---|---|---|---|---|---|---|---|---|
| | | CNLI | HeSw | PIQA | CHID | WSC$_P$ | WSC$_G$ | CSQA | BoolQ | MMLU | CMLU | Race$_H$ | Race$_M$ | XSum | C3 | |
| Base | 0 | 32.98 | 71.34 | 78.18 | 41.56 | 37.50 | 38.46 | 55.04 | 70.70 | 46.67 | 31.88 | 35.53 | 33.36 | 19.55 | 43.84 | 45.47 |
| Base | 12.5 | 32.99 | 67.06 | 74.92 | 39.61 | 36.53 | 1.92 | 57.41 | 69.36 | 47.15 | 31.61 | 39.11 | 38.65 | 17.59 | 44.60 | 42.75 |
| Base | 25 | 32.98 | 63.80 | 69.21 | 35.37 | 36.54 | 0.00 | 50.78 | 64.74 | 40.80 | 30.31 | 35.19 | 35.62 | 16.11 | 43.51 | 39.64 |
| Base | 37.5 | 32.58 | 45.04 | 61.53 | 20.68 | 36.54 | 2.88 | 42.18 | 64.43 | 39.87 | 29.42 | 31.90 | 29.74 | 2.77 | 41.37 | 34.35 |
| Base | 50 | 34.51 | 34.89 | 55.33 | 17.08 | 36.54 | 11.54 | 19.82 | 62.29 | 28.72 | 25.10 | 23.41 | 26.04 | 1.21 | 35.07 | 29.40 |
| Base | 62.5 | 35.14 | 29.71 | 52.83 | 14.94 | 39.42 | 1.92 | 21.46 | 50.06 | 24.55 | 25.16 | 26.76 | 25.42 | 0.09 | 27.62 | 26.80 |
| Base | 75 | 34.94 | 26.71 | 51.03 | 13.59 | 36.54 | 8.65 | 20.56 | 52.60 | 24.23 | 24.47 | 23.18 | 22.63 | 0.08 | 27.29 | 26.17 |
| LM | 0 | 31.30 | 71.28 | 75.95 | 36.11 | 63.46 | 59.62 | 64.29 | 74.77 | 48.30 | 33.93 | 52.52 | 55.22 | 22.45 | 47.56 | 52.63 |
| LM | 12.5 | 32.42 | 67.58 | 72.72 | 28.91 | 50.92 | 60.50 | 60.92 | 72.88 | 46.69 | 32.02 | 51.34 | 54.45 | 18.26 | 45.94 | 49.68 |
| LM | 25 | 30.10 | 60.63 | 66.82 | 20.53 | 48.96 | 42.31 | 65.88 | 70.82 | 42.09 | 32.40 | 48.23 | 50.43 | 15.75 | 43.62 | 45.11 |
| LM | 37.5 | 33.29 | 45.13 | 60.66 | 20.03 | 36.54 | 11.73 | 59.38 | 68.07 | 39.18 | 29.64 | 39.71 | 42.20 | 6.36 | 41.04 | 39.40 |
| LM | 50 | 34.93 | 34.67 | 56.20 | 16.18 | 36.54 | 8.65 | 22.28 | 62.14 | 32.01 | 26.44 | 25.39 | 25.49 | 2.34 | 35.01 | 29.88 |
| LM | 62.5 | 34.11 | 30.50 | 53.21 | 14.34 | 51.92 | 2.88 | 20.56 | 57.95 | 24.58 | 25.21 | 23.13 | 23.75 | 0.18 | 27.12 | 27.82 |
| LM | 75 | 34.87 | 27.03 | 52.19 | 14.54 | 39.42 | 0.00 | 20.23 | 53.87 | 24.45 | 24.83 | 21.41 | 22.14 | 0.02 | 26.69 | 25.82 |
| Math | 0 | 32.99 | 68.60 | 75.79 | 39.71 | 39.42 | 36.54 | 50.78 | 69.36 | 43.04 | 32.16 | 30.36 | 36.42 | 20.88 | 43.45 | 44.25 |
| Math | 12.5 | 32.97 | 64.72 | 73.06 | 37.50 | 23.08 | 23.07 | 51.43 | 71.16 | 42.91 | 31.90 | 32.99 | 36.07 | 19.30 | 43.83 | 41.71 |
| Math | 25 | 34.92 | 46.24 | 61.92 | 19.38 | 36.54 | 56.73 | 45.45 | 72.81 | 35.07 | 29.78 | 31.45 | 34.33 | 6.24 | 39.89 | 39.34 |
| Math | 37.5 | 32.99 | 55.42 | 62.81 | 23.82 | 38.38 | 4.81 | 37.87 | 68.68 | 36.46 | 27.19 | 28.02 | 33.79 | 13.88 | 39.37 | 36.04 |
| Math | 50 | 32.73 | 35.93 | 55.06 | 16.73 | 39.42 | 39.42 | 20.15 | 64.34 | 29.94 | 25.52 | 26.82 | 26.60 | 2.31 | 35.56 | 32.15 |
| Math | 62.5 | 34.93 | 31.06 | 54.08 | 13.79 | 58.65 | 4.81 | 20.56 | 46.24 | 26.70 | 25.05 | 26.56 | 26.53 | 0.57 | 28.33 | 28.42 |
| Math | 75 | 34.94 | 27.35 | 52.07 | 14.39 | 43.27 | 2.88 | 20.88 | 56.51 | 24.25 | 23.14 | 24.76 | 24.79 | 0.15 | 27.45 | 27.20 |
| Code | 0 | 32.99 | 70.27 | 78.62 | 41.61 | 36.54 | 41.35 | 57.41 | 71.04 | 46.22 | 32.20 | 41.25 | 39.69 | 18.79 | 46.25 | 46.73 |
| Code | 12.5 | 32.97 | 65.79 | 75.78 | 39.06 | 36.54 | 0.96 | 56.67 | 71.13 | 47.09 | 32.00 | 44.73 | 44.84 | 19.21 | 47.29 | 43.86 |
| Code | 25 | 32.99 | 63.06 | 72.02 | 35.67 | 36.54 | 0.00 | 50.59 | 68.87 | 40.50 | 28.87 | 36.64 | 38.59 | 17.59 | 45.64 | 40.51 |
| Code | 37.5 | 33.21 | 44.12 | 62.13 | 20.78 | 36.54 | 2.88 | 48.81 | 63.91 | 40.29 | 29.56 | 36.25 | 35.52 | 5.35 | 42.14 | 35.82 |
| Code | 50 | 34.93 | 34.15 | 54.95 | 16.73 | 36.54 | 17.31 | 22.03 | 62.54 | 28.46 | 25.16 | 24.13 | 24.44 | 2.03 | 36.62 | 30.00 |
| Code | 62.5 | 34.72 | 29.67 | 52.99 | 14.39 | 40.38 | 8.65 | 22.52 | 50.70 | 24.78 | 25.15 | 27.16 | 28.04 | 0.12 | 27.78 | 27.50 |
| Code | 75 | 34.94 | 26.79 | 50.82 | 13.99 | 38.46 | 5.77 | 24.08 | 48.38 | 24.08 | 24.52 | 22.73 | 22.49 | 0.13 | 27.29 | 26.03 |
| Ours | 0 | 36.88 | 73.16 | 78.67 | 39.46 | 64.46 | 45.19 | 65.37 | 78.43 | 49.75 | 35.08 | 58.78 | 61.65 | 24.50 | 49.33 | 54.34 |
| Ours | 12.5 | 33.00 | 66.78 | 75.19 | 34.92 | 64.42 | 63.46 | 63.98 | 75.87 | 48.79 | 34.13 | 53.89 | 56.20 | 20.21 | 45.37 | 52.59 |
| Ours | 25 | 32.99 | 57.31 | 68.34 | 22.38 | 63.46 | 63.46 | 57.58 | 62.17 | 45.92 | 30.96 | 52.20 | 56.06 | 7.12 | 39.67 | 47.11 |
| Ours | 37.5 | 35.67 | 51.02 | 63.44 | 20.68 | 62.50 | 22.00 | 57.99 | 67.52 | 47.09 | 34.11 | 44.00 | 46.38 | 2.96 | 39.34 | 42.00 |
| Ours | 50 | 33.97 | 41.99 | 58.16 | 21.08 | 38.54 | 24.12 | 26.52 | 46.03 | 32.32 | 28.30 | 28.99 | 28.88 | 6.30 | 36.11 | 32.23 |
| Ours | 62.5 | 33.30 | 28.34 | 51.96 | 18.09 | 46.15 | 6.88 | 23.88 | 45.81 | 26.41 | 26.95 | 28.73 | 28.72 | 5.09 | 28.47 | 28.48 |
| Ours | 75 | 34.93 | 30.45 | 49.18 | 20.48 | 39.54 | 10.81 | 21.98 | 45.29 | 25.28 | 24.68 | 26.30 | 26.93 | 0.46 | 28.38 | 27.47 |

**Post-training Setup.** We selected two competitive baseline methods (ShortGPT, LACO) and followed the recovery-phase setting from LLM-Pruner. We used the cleaned Alpaca dataset (50k samples) and fine-tuned with the LoRA configuration: rank (d=8), learning rate = 1e-4, 100 warm-up steps, batch size = 64, AdamW optimizer, and 2 training epochs.

**Computational cost scaling with candidate number.** The computational cost increases with the number of candidate models, primarily due to the need for longer search trails to ensure we find optimal performance points. As shown in the table Table 17.

## H.4  SCALING TO STRONGER THINKING MODEL

We further extend our method to recent thinking models. Specifically, we evaluate our approach using Qwen3-4B-Instruct (LM) and Qwen3-4B-Thinking (Thinking) models (Team, 2025). The results are presented in Table 19, demonstrating the effectiveness of our method on this emerging model architecture.

Table 11: The main results of the Llama3-8B model across multiple natural language benchmarks using candidate models: Meta-Llama-3-8B-Instruct (LM), MathCoder2-Llama-3-8B (Math), Code-Llama-3-8B (Code), and Meta-Llama-3-8B (Base). "PTM" (Pruning-then-Merging) refers to first pruning each candidate model using the current pruner and then merging them. "MTP" (Merging-then-Pruning) refers to first merging the candidate models and then applying pruning.

| LLM | Pruner ratio/layer | Type | Reasoning | | | Language | | | Knowledge | | | | Understanding | | | | Avg |
|---|---|---|---|---|---|---|---|---|---|---|---|---|---|---|---|---|---|
| | | | CMNLI | HeSw | PIQA | CHID | $WSC_P$ | $WSC_G$ | CSQA | BoolQ | MMLU | CMLU | $Race_H$ | $Race_M$ | XSum | C3 | |
| Llama3 -8B | Dense | Base | 32.98 | 74.67 | 80.96 | 73.78 | 56.73 | 36.54 | 73.79 | 69.97 | 64.74 | 50.79 | 63.21 | 70.54 | 3.28 | 55.18 | 57.65 |
| | | LM | 33.00 | 71.08 | 80.69 | 65.53 | 55.77 | 69.23 | 76.66 | 78.87 | 65.97 | 53.64 | 76.44 | 81.75 | 17.97 | 63.95 | 63.61 |
| | | Math | 32.99 | 71.66 | 77.97 | 57.09 | 37.50 | 58.65 | 68.22 | 69.08 | 62.08 | 45.85 | 64.75 | 69.08 | 8.68 | 53.86 | 55.53 |
| | | Code | 32.98 | 65.56 | 74.70 | 78.42 | 61.54 | 61.54 | 63.47 | 78.35 | 48.03 | 34.55 | 52.40 | 58.43 | 19.36 | 46.41 | 55.41 |
| | ShortGPT (24.6%) | Base | 36.00 | 31.36 | 62.84 | 25.77 | 36.54 | 63.46 | 53.97 | 50.61 | 36.05 | 33.83 | 30.73 | 32.38 | 1.17 | 38.96 | 38.12 |
| | | LM | 32.83 | 45.06 | 65.78 | 23.38 | 41.35 | 53.85 | 39.56 | 63.73 | 32.37 | 28.69 | 40.14 | 45.19 | 3.68 | 43.51 | 39.94 |
| | | Math | 32.98 | 42.89 | 63.00 | 17.18 | 36.54 | 36.54 | 45.37 | 46.30 | 33.95 | 29.71 | 28.87 | 30.22 | 1.45 | 40.49 | 34.68 |
| | | Code | 32.26 | 45.99 | 64.96 | 17.03 | 36.54 | 36.54 | 36.20 | 63.98 | 28.78 | 26.25 | 27.27 | 29.46 | 3.57 | 39.01 | 34.85 |
| | | MTP | 32.98 | 48.51 | 64.85 | 18.33 | 36.54 | 35.58 | 42.83 | 67.06 | 33.05 | 28.73 | 30.07 | 32.66 | 3.64 | 44.33 | 37.08 |
| | | PTM | 32.95 | 48.58 | 64.96 | 18.43 | 36.54 | 35.58 | 42.83 | 67.22 | 33.05 | 28.71 | 30.16 | 32.45 | 3.66 | 44.27 | 37.10 |

Table 12: Architecture Parameters of pruned 13B models

| Layer | Model-1 | | | Model-2 | | | Model-3 | | |
|---|---|---|---|---|---|---|---|---|---|
| | Type | Merge Factor | Output Scale | Type | Merge Factor | Output Scale | Type | Merge Factor | Output Scale |
| 0 | Base | - | 1.00 | LM | - | 1.00 | LM | - | 1.00 |
| 1 | LM | - | 1.00 | LM+Math | 0.64 | 1.00 | Base | - | 1.00 |
| 2 | LM | - | 1.00 | LM+Code | 0.60 | 1.05 | LM+Code | 0.60 | 1.05 |
| 3 | LM | - | 1.00 | LM | - | 1.00 | LM+Code | 0.60 | 1.00 |
| 4 | LM | - | 1.00 | LM | - | 1.00 | LM | - | 1.00 |
| 5 | Code | - | 1.00 | LM+Math | 0.59 | 1.00 | LM+Math | 0.58 | 1.00 |
| 6 | Base | - | 1.00 | LM | - | 1.00 | LM | - | 1.00 |
| 7 | LM | - | 1.00 | LM+Math | 0.60 | 1.00 | LM+Math | 0.60 | 1.00 |
| 8 | LM | - | 1.00 | LM | - | 1.00 | LM+Code | 0.59 | 1.00 |
| 9 | LM | - | 1.00 | LM | - | 0.84 | LM | - | 0.93 |
| 10 | LM | - | 1.00 | LM | - | 1.02 | LM | - | 1.22 |
| 11 | LM | - | 1.00 | LM+Code | 0.66 | 0.77 | LM+Math | 0.66 | 1.00 |
| 12 | LM | - | 0.91 | LM+Code | 0.60 | 1.00 | LM+Code | 0.60 | 1.13 |
| 13 | LM+Code | 0.70 | 1.00 | LM+Math | 0.60 | 1.00 | LM+Math +Code | 0.60 | 1.11 |
| 14 | LM+Math | 0.70 | 1.00 | LM+Math | 0.60 | 1.00 | LM | - | 1.00 |
| 15 | LM | - | 1.00 | LM+Math | 0.70 | 1.00 | LM+Math | 0.66 | 1.00 |
| 16 | Base | - | 1.00 | LM+Math | 0.60 | 1.00 | LM+Math | 0.60 | 1.00 |
| 17 | LM | - | 1.00 | LM | - | 1.00 | LM | - | 1.00 |
| 18 | LM | - | 1.00 | REMOVED | | | REMOVED | | |
| 19 | LM+Code | 0.70 | 1.00 | LM+Code | 0.60 | 1.00 | LM+Code | 0.60 | 1.01 |
| 20 | LM+Code | 0.70 | 1.00 | LM | - | 1.00 | REMOVED | | |
| 21 | LM | - | 1.00 | Base | - | 1.07 | Base | - | 1.07 |
| 22 | LM | - | 1.00 | Math | - | 1.00 | LM+Math | 0.60 | 1.09 |
| 23 | LM | - | 1.00 | REMOVED | | | REMOVED | | |
| 24 | LM | - | 1.00 | Base | - | 1.01 | Base | - | 1.01 |
| 25 | REMOVED | | | REMOVED | | | REMOVED | | |
| 26 | REMOVED | | | LM | - | 1.04 | LM | - | 1.04 |
| 27 | REMOVED | | | REMOVED | | | REMOVED | | |
| 28 | REMOVED | | | REMOVED | | | REMOVED | | |
| 29 | REMOVED | | | REMOVED | | | REMOVED | | |
| 30 | REMOVED | | | Base | - | 1.00 | Base | - | 1.00 |
| 31 | REMOVED | | | REMOVED | | | REMOVED | | |
| 32 | REMOVED | | | REMOVED | | | LM | - | 1.00 |
| 33 | REMOVED | | | REMOVED | | | REMOVED | | |
| 34 | LM | - | 1.00 | Base | - | 1.00 | Code | - | 1.00 |
| 35 | Base | - | 1.00 | LM | - | 1.13 | LM | - | 1.28 |
| 36 | LM | - | 1.00 | REMOVED | | | REMOVED | | |
| 37 | LM | - | 1.00 | LM | - | 1.00 | LM | - | 1.00 |
| 38 | LM | - | 0.75 | LM | - | 1.00 | Math | - | 1.00 |
| 39 | REMOVED | | | Math | - | 1.00 | Math | - | 1.00 |

## H.5 SCALING TO MATH AND CODE TASKS

We conducted additional experiments on mathematical and coding tasks using LLaMA-7B, comparing our approach with the two strongest baseline methods (ShortGPT and LACO) under varying numbers of pruned layers. As shown in the Table 20, tasks that require structured output formats, such as mathematical reasoning and code generation, are particularly sensitive to layer removal. The baseline methods exhibit catastrophic drops in performance, with the removal of just 2-4 layers leading to

Table 13: Architecture Parameters of pruned 7B models

| Layer | Model-1 | | | Model-2 | | | Model-3 | | |
| | Type | Merge Factor | Output Scale | Type | Merge Factor | Output Scale | Type | Merge Factor | Output Scale |
|---|---|---|---|---|---|---|---|---|---|
| 0 | LM | - | 1.00 | Math+Code | 0.48 | 1.00 | LM+Math | 0.48 | 0.92 |
| 1 | LM+Math+Code | 0.50 | 1.00 | LM | - | 1.00 | LM | - | 1.00 |
| 2 | LM | - | 1.03 | LM+Code | 0.52 | 1.06 | LM | - | 1.03 |
| 3 | LM | - | 1.00 | Base | - | 0.98 | Math | - | 1.05 |
| 4 | LM | - | 1.04 | LM | - | 1.11 | LM | - | 1.11 |
| 5 | LM+Code | 0.59 | 1.08 | LM+Math | 0.38 | 1.12 | LM | - | 1.13 |
| 6 | Code | - | 1.19 | Math | - | 1.25 | Code | - | 1.11 |
| 7 | Code | - | 0.88 | LM+Code | 0.50 | 0.77 | LM+Code | 0.50 | 0.77 |
| 8 | LM | - | 1.28 | LM | - | 1.34 | LM | - | 1.19 |
| 9 | LM | - | 0.86 | LM | - | 0.93 | LM+Code | 0.51 | 0.56 |
| 10 | Base | - | 1.00 | LM | - | 1.00 | LM | - | 1.00 |
| 11 | LM+Math | 0.50 | 1.00 | Math | - | 1.02 | LM | - | 1.05 |
| 12 | LM | - | 1.00 | LM+Math | 0.41 | 0.99 | LM+Math | 0.41 | 1.00 |
| 13 | Math | - | 1.00 | LM+Math | 0.50 | 1.20 | LM+Math | 0.58 | 1.20 |
| 14 | LM+Math | 0.60 | 1.00 | LM | - | 1.00 | LM+Math | 0.54 | 1.00 |
| 15 | LM | - | 1.18 | Code | - | 0.97 | Code | - | 1.05 |
| 16 | LM+Math | 0.50 | 1.00 | LM+Math | 0.50 | 1.00 | LM+Math | 0.45 | 1.00 |
| 17 | LM+Math+Code | 0.50 | 1.00 | Code | - | 1.00 | Math+Code | 0.50 | 1.00 |
| 18 | Math+Code | 0.50 | 1.00 | Base | - | 1.00 | Base | - | 1.01 |
| 19 | REMOVED | | | REMOVED | | | REMOVED | | |
| 20 | REMOVED | | | REMOVED | | | REMOVED | | |
| 21 | LM | - | 1.00 | REMOVED | | | LM | - | 1.00 |
| 22 | REMOVED | | | REMOVED | | | REMOVED | | |
| 23 | REMOVED | | | REMOVED | | | REMOVED | | |
| 24 | REMOVED | | | LM | - | 1.00 | REMOVED | | |
| 25 | REMOVED | | | REMOVED | | | REMOVED | | |
| 26 | REMOVED | | | REMOVED | | | REMOVED | | |
| 27 | LM | - | 1.00 | Base | - | 0.99 | LM | - | 0.99 |
| 28 | REMOVED | | | LM | - | 1.00 | REMOVED | | |
| 29 | LM+Code | 0.50 | 1.00 | LM | - | 1.00 | LM+Code | 0.50 | 1.00 |
| 30 | REMOVED | | | REMOVED | | | REMOVED | | |
| 31 | LM+Math | 0.50 | 1.00 | REMOVED | | | LM+Math | 0.50 | 1.00 |

Table 14: Comparison of different searching settings across various benchmarks. Settings: LR-only: Layer-remove only, LS+LR: Layer-selection + layer-remove, FL-merge: Folding Layers Merging.

| Setting | Reasoning | | | Language | | | Knowledge | | | | Understanding | | | | Avg |
| | CNLI | HeSw | PIQA | CHID | WSC$_P$ | WSC$_G$ | CSQA | BoolQ | MMLU | CMLU | Race$_H$ | Race$_M$ | XSum | C3 | |
|---|---|---|---|---|---|---|---|---|---|---|---|---|---|---|---|
| Ours | 35.46 | 54.43 | 67.74 | 23.63 | 63.46 | 43.27 | 62.90 | 75.08 | 48.75 | 33.86 | 55.35 | 58.64 | 12.99 | 44.16 | 48.55 |
| LR-only | 34.96 | 53.80 | 66.70 | 18.58 | 49.04 | 58.65 | 60.61 | 68.87 | 47.85 | 33.54 | 42.51 | 43.04 | 8.05 | 41.42 | 44.83 |
| LS+LR | 32.92 | 55.84 | 65.07 | 17.98 | 63.46 | 26.92 | 58.97 | 51.22 | 48.97 | 34.61 | 48.68 | 49.44 | 8.33 | 42.41 | 43.20 |
| FL-merge | 32.99 | 52.90 | 63.66 | 19.28 | 46.15 | 62.50 | 60.52 | 75.20 | 48.30 | 34.33 | 50.77 | 55.29 | 6.39 | 39.40 | 46.26 |

near-zero performance. In contrast, our method **consistently maintains superior performance across all pruning ratios**.

Table 15: Budget allocation to search trials for pruning. $41\%$ of trials require only the smallest budget size, significantly increasing efficiency.

| Dataset | Low Budget (**41.4%, 207 trials**) | Medium Budget (36.6%, 183 trials) | High Budget (22.0%, 110 trials) |
|---------|---------|---------|---------|
| PIQA | 100 | 300 | 1000 |
| WSC | 100 | 200 | 500 |
| CSQA | 100 | 300 | 1000 |
| MMLU | 100 | 300 | 1000 |

Table 16: Evaluation runtime (Seconds) different datasets and sample sizes

| Dataset | Size | Runtime (S) |
|---------|------|-------------|
| CSQA | 100 | 2.76 |
| | 300 | 6.97 |
| | 1000 | 16.51 |
| WSC | 100 | 2.41 |
| | 200 | 2.50 |
| | 500 | 2.67 |
| PIQA | 100 | 2.75 |
| | 300 | 7.00 |
| | 1000 | 21.23 |
| MMLU | 100 | 2.56 |
| | 300 | 6.49 |
| | 1000 | 21.66 |

Table 17: Scaling of Computational Cost with Number of Models

| Number of Models | Search Trials | FLOPs |
|------------------|---------------|-------|
| 1 | 200 | $9.85 \times 10^{15}$ |
| 2 | 300 | $2.26 \times 10^{16}$ |
| 3 | 500 | $9.35 \times 10^{16}$ |

Table 18: Comparison of efficiency of pruning methods

| Metric | LACO | ShortGPT | Ours (Multi-models) | Ours (Layer Folding) |
|--------|------|----------|---------------------|----------------------|
| **Pruning Stage** | | | | |
| FLOPS | 1.29e+14 | 4.91e+19 | 9.35e+16 | 1.75e+16 |
| Performance (avg) | 37.14 | 42.40 | 48.55 | 46.26 |
| **Post-training Stage** | | | | |
| FLOPS | 1.06e+18 | 1.06e+18 | 0 | 0 |
| Performance (avg) | 40.03 | 42.76 | 48.55 | 46.26 |
| **Overall Summary** | | | | |
| Total FLOPS | 1.06e+18 | 4.91e+19 | 9.35e+16 | 1.75e+16 |
| Final Accuracy | 40.03 | 42.76 | **48.55** | **46.26** |

Table 19: Comparison of pruning methods on multiple natural language benchmarks. "Single" refers to the best performance achieved when pruning a single model directly, while "Merge" refers to the best performance achieved through either "pruning-then-merging" or "merging-then-pruning". 4B models: Qwen3-4B-Instruct (LM), Qwen3-4B-Thinking (Thinking).

| LLM | Pruner ratio/layer | Type | Reasoning | | | Language | | | Knowledge | | | | Understanding | | | | Avg |
|---|---|---|---|---|---|---|---|---|---|---|---|---|---|---|---|---|---|
| | | | CMNLI | HeSw | PIQA | CHID | $WSC_P$ | $WSC_G$ | CSQA | BoolQ | MMLU | CMLU | $Race_H$ | $Race_M$ | XSum | C3 | |
| Qwen3-4B | Dense | Base | 38.83 | 64.20 | 75.68 | 79.67 | 48.08 | 55.77 | 80.34 | 80.37 | 72.43 | 73.52 | 65.95 | 73.33 | 14.73 | 67.62 | 63.61 |
| | | Thinking | 45.22 | 60.60 | 75.52 | 79.02 | 62.50 | 65.38 | 77.81 | 82.45 | 70.57 | 71.85 | 69.73 | 78.13 | 1.78 | 67.73 | 64.88 |
| | ShortGPT | Single | 35.45 | 44.78 | 67.03 | 53.55 | 63.46 | 30.77 | 49.63 | 63.39 | 44.40 | 46.09 | 35.96 | 39.21 | **12.56** | 51.95 | 45.59 |
| | | Merge | 33.09 | 43.18 | 67.36 | 52.65 | 60.58 | 20.60 | 32.76 | 63.33 | 32.30 | 32.16 | 30.93 | 28.34 | 11.34 | 49.81 | 39.89 |
| | Ours | | **36.07** | **45.94** | **68.39** | **56.29** | **64.42** | **35.60** | **62.00** | **67.71** | **48.56** | **47.04** | **37.85** | **40.81** | 10.20 | **52.55** | 48.10 |

Table 20: Performance comparison on mathematical and coding tasks across different pruning ratios using LLaMA-7B.

| Method | Layers Pruned | GSM8K | HumanEval |
|---|---|---|---|
| **Candidate Models (No Pruning)** | | | |
| base | 0 | 11.30 | 3.05 |
| lm | 0 | 21.23 | 3.05 |
| math | 0 | 11.99 | 0.00 |
| code | 0 | 3.11 | 14.02 |
| LACO | 2 | 3.80 | 6.71 |
| | 4 | 0.76 | 1.22 |
| | 6 | 0.00 | 0.00 |
| | 8 | 0.00 | 0.00 |
| ShortGPT | 2 | 1.50 | 2.44 |
| | 4 | 0.00 | 0.61 |
| | 6 | 0.00 | 0.00 |
| | 8 | 0.00 | 0.00 |
| Ours | 2 | **22.22** | **12.81** |
| | 4 | **15.24** | **6.10** |
| | 6 | **5.31** | **1.22** |
| | 8 | 0.00 | 0.00 |

Table 21: Robustness analysis of candidate model combinations across multiple natural language benchmarks. Blue-highlighted cells show optimal performance using three specialized models: Llama-2-7B-Chat (LM), MAmmoTH-7B (Math), and Llama-2-Coder-7B (Code), with Llama-2-7B serving as the base model.

| Model Pool | Reasoning | | | Language | | | Knowledge | | | | Understanding | | | | Avg |
|---|---|---|---|---|---|---|---|---|---|---|---|---|---|---|---|
| | CMNLI | HeSw | PIQA | CHID | WSCP | WSCG | CSQA | BoolQ | MMLU | CMMLU | RaceH | RaceM | XSum | C3 | |
| **3-candidate models** | | | | | | | | | | | | | | | |
| Math+LM+Code | **35.46** | 54.43 | 67.74 | 23.63 | 63.46 | 43.27 | 62.90 | 75.08 | 48.75 | 33.86 | 55.35 | 58.64 | 12.99 | 44.16 | **48.55** |
| **2-candidate models** | | | | | | | | | | | | | | | |
| Math+LM | 32.93 | 55.93 | 67.90 | 20.93 | 57.69 | 57.69 | 62.24 | 76.54 | 45.31 | 33.25 | 49.06 | 53.27 | 14.42 | 42.30 | 47.82 |
| Code+LM | 33.00 | 58.09 | 67.52 | 21.08 | 56.73 | 50.96 | 62.65 | 70.09 | 46.96 | 33.85 | 50.31 | 55.36 | 12.38 | 43.40 | 47.31 |
| Code+Math | 32.93 | 53.67 | 69.53 | 27.27 | 38.46 | 34.60 | 56.35 | 65.99 | 41.40 | 30.97 | 45.14 | 44.43 | 8.11 | 44.60 | 43.12 |
| **1-candidate model** | | | | | | | | | | | | | | | |
| LM | 33.27 | 51.34 | 64.20 | 19.33 | 62.50 | 53.85 | 62.82 | 64.86 | 46.47 | 31.59 | 47.80 | 51.39 | 6.97 | 39.51 | 45.42 |
| Math | 32.95 | 60.65 | 66.49 | 22.43 | 36.54 | 32.50 | 58.07 | 71.01 | 44.13 | 32.07 | 40.28 | 41.57 | 13.76 | 41.10 | 42.40 |
| Code | 30.10 | 54.72 | 69.75 | 26.17 | 63.46 | 62.50 | 50.94 | 65.00 | 36.42 | 26.69 | 31.02 | 30.99 | 2.47 | 38.14 | 42.03 |
| **base only** | | | | | | | | | | | | | | | |
| Base | 32.22 | 56.48 | 67.46 | 26.32 | 61.54 | 50.00 | 41.44 | 66.91 | 40.54 | 28.01 | 37.94 | 39.35 | 0.96 | 41.92 | 42.20 |

## H.6 SCALING TO OTHER CANDIDATE MODELS

To clarify the role of domain diversity in candidate model selection, we note that strict domain diversity is not always necessary. The optimal combination depends on the optimization objective: if the goal is improving performance on language tasks, including more high-quality language models in the candidate pool is naturally beneficial. However, when access to same-domain models is limited, a diverse candidate pool can still provide comparable results through complementary capabilities. To validate this, we conducted an additional experiment using a candidate pool with only two models: a Llama-7b instruct model(LM) and a Chinese fine-tuned Llama-7b model(CN_LM). As shown in Table 22, This focused selection of high-quality language models achieved even better performance than our main results, confirming that strategic model selection can be more effective than broad diversity when models are well-aligned with the target task.

Table 22: Comparison with other candidate models using high-quality language models. The experiment shows that using two specialized language models (LM and CN_LM) can achieve superior performance.

| Method | Reasoning | | | Language | | | Knowledge | | | | Understanding | | | | Avg |
|---|---|---|---|---|---|---|---|---|---|---|---|---|---|---|---|
| | CMNLI | HeSw | PIQA | CHID | WSCP | WSCG | CSQA | BoolQ | MMLU | CMMLU | RaceH | RaceM | XSum | C3 | |
| Base | 32.98 | 71.34 | 78.18 | 41.56 | 37.50 | 38.46 | 55.04 | 70.70 | 46.67 | 31.88 | 35.53 | 33.36 | 19.55 | 43.84 | 45.47 |
| CN_LM | 34.02 | 70.03 | 76.71 | 38.31 | 63.46 | 59.62 | 61.51 | 56.09 | 46.47 | 32.64 | 41.48 | 45.47 | 17.64 | 46.58 | 49.29 |
| LM | 31.30 | 71.28 | 75.95 | 36.11 | 63.46 | 59.62 | 64.29 | 74.77 | 48.30 | 33.93 | 52.52 | 55.22 | 22.45 | 47.56 | 52.63 |
| ShortGPT | 34.14 | 33.74 | 59.85 | 15.23 | 61.54 | 33.46 | 44.81 | 55.20 | 30.70 | 27.06 | 40.73 | 42.78 | 13.20 | 34.58 | 37.64 |
| Ours | 33.00 | 63.24 | 68.00 | 22.43 | 60.69 | 57.69 | 63.64 | **76.02** | 45.31 | 33.25 | **50.08** | **53.30** | 14.42 | 42.26 | **48.80** |

