# OpenReview forum: "GPTailor: Large Language Model Pruning Through Layer Cutting and Stitching"
_ICLR.cc/2026/Conference — ICLR 2026 Poster_

### Official Review · Reviewer_unTV · 2025-10-28

**Soundness:** 3
**Presentation:** 3
**Contribution:** 3
**Rating:** 6
**Confidence:** 4

**Summary:**

This paper presents a novel and interesting approach to structured pruning by leveraging multiple fine-tuned models for layer selection, removal, and merging. It breaks away from the traditional single-model pruning framework and attempts to construct a smaller but more comprehensive model by stitching the expertise of different models. The paper conduct extensive experiments on the Llama2-7B and 13B models and demonstrate its effectiveness.

**Strengths:**

- The idea of ​​compressing a model by strategically combining or merging layers from fine-tuned model variants is novel.
- The ablation experiment is very thorough

**Weaknesses:**

- The proposed method is not tested on a larger 70B model, so its generalization cannot be demonstrated.
- The algorithm part is very vague.
- Lack of theoretical support, why can better results be achieved by stitching multiple models?
- There are too few baselines for comparison. Please compare more baselines, such as [1-3].
- Lack of ablation experiments on Calibration Data.
- As your mention in sec3.2, If $\sum^K_{j=1} c_{i,j} = 0$, we retain the layer from the base model instead. If $\sum^K_{j=1} c_{i,j} = 0$, why not remove this layer?
- The author only gives a general description of how to optimize the search space to obtain the final pruned model. This part seems more like the result of manual attempts, because the search space is large when using a search algorithm, and how to ensure that the model you searched is the global optimal one. I hope the author can give a detailed and clear answer.

[1] Reassessing layer pruning in llms: New insights and methods. arXiv preprint arXiv:2411.15558

[2] Streamlining redundant layers to compress large language models. arXiv preprint arXiv:2403.19135.

[3] Finercut: Finer-grained interpretable layer pruning for large language models. arXiv preprint arXiv:2405.18218.

**Questions:**

- If the performance of the original model you choose is too poor, will it affect the merged or spliced ​​pruned model?
- How much impact will switching to other merge methods have on the performance of the pruned model?

---

> ### Author Response · Authors · 2025-11-20
>
> Thank you for for your detailed feedback. We’re encouraged that you considered our approach novel and our experimental design comprehensive. We’ll answer your questions below.
>
> ## The presentation of the algorithm
>
> Thank you for raising this concern. We acknowledge that space constraints limited our algorithm presentation in the main text. To address this, we have now added more detailed implementation specifics of SMAC in **Appendix E**. We would be happy to provide additional clarifications. Please let us know which specific parts require further explanation, and we will gladly include more detailed descriptions.
>
> ## Comparison with other Baselines
>
> Thank you for bringing these papers to our attention. We now compare our method with the two best-performing pruning approaches: Reverse-order in [3] and STREAMLINING_llm [4], following the evaluation benchmark from [4] on LLaMA-7B. As shown in the table below, our method achieves 50.27% average accuracy, outperforming Reverse-order and STREAMLINING_llm.
>
>
> | Method | C3 | CMNLI | CHID | BoolQ | WSC | CoQA | HeSWPI | QARC | Race-M | Race-H | MMLU | CMMLU | Avg |
> |--------|-----|-------|------|--------|-----|------|--------|------|--------|--------|------|-------|-------|
> | Reverse-order [3] | 40.22 | 32.86 | 20.03 | 68.21 | 42.31 | 61.67 | 54.69 | 62.46 | 40.65 | 34.95 | 47.40 | 32.91 | 44.86 |
> | STREAMLINING_llm [4] | 40.73 | 32.00 | 22.86 | 5.93 | 8.50 | 60.60 | 61.20 | 71.20 | 38.00 | 38.70 | 47.00 | 31.70 | 45.78 |
> | Ours | 44.16 | 35.46 | 23.63 | 75.08 | 43.27 | 62.9 | 54.43 | 67.74 | 58.64 | 55.35 | 48.75 | 33.86 | 50.27 |
>
>
> ## Theoretical support
>
>  We have updated the PDF to provide deeper analysis (**Appendix F**) from three perspectives: the theoretical foundation of model merging, empirical observations of layer-wise patterns, and post-hoc analysis of layer characteristics. Please take a look at the updated PDF to find out more.
>
>
> ## Regarding The Design Choices in Section 3.2
> The design decouples layer removal (controlled by **r** to enforce sparsity) from layer source selection (controlled by **c**), where ∑cᵢ,ⱼ = 0 means "none of the fine-tuned candidates are beneficial, so fall back to the base model layer" rather than "this layer should be removed," which maintains the sparsity constraint and allows more exploration of which positions benefit from fine-tuned layers versus retaining the original base model.
>
>
> ## Response to Search Space and Optimization
>
> Thank you for this thoughtful question. We'd like to clarify our search space design and optimization guarantees:
>
> **Conceptual Simplicity:** Our search space is intentionally **simple and straightforward** and constructed by first principles, instead of being the result of manual tuning. Our search space support three basic operations: layer pruning, layer selection, and layer merging, with a single simple merging algorithm. Arguably, implementing pruning and selection as binary variables and merging as search over the parameters of a standard merging method, is the most principled search space covering these operations. Despite this simplicity, we already achieve strong results. This suggests our approach is robust and not overly dependent on manual design.
>
>
>
>
> **How SMAC finds high-quality solutions**: SMAC combines Bayesian optimization with Hyperband-style early stopping. It builds a probabilistic surrogate model to predict configuration performance, uses an acquisition function to balance exploration and exploitation, and applies early stopping to terminate unpromising configurations efficiently. The surrogate model is adaptively updated after each evaluation, progressively focusing the search on promising regions of the configuration space.
>
> **On global optimality**: We acknowledge that SMAC cannot guarantee finding the global optimum, as is the case with any practical optimization method on such large discrete search spaces. However, Bayesian optimization methods like SMAC provide theoretical convergence guarantees: given sufficient evaluations, they converge to near-optimal solutions. In practice, with limited computational budgets, SMAC efficiently identifies high-quality solutions that significantly outperform baseline methods.
>
> **Empirical evidence of convergence**: The table below shows the optimization trajectory on LLaMA-7B over 500 trials. The search progressively converges toward better solutions, identifying strong configurations around iteration 300 despite running 500 trials total. This demonstrates both the convergence behavior and SMAC's ability to find high-quality solutions within practical computational budgets.
>
> | Iteration | Accuracy (%) |
> |-----------|--------------|
> | 16        | 32.67        |
> | 29        | 34.21        |
> | 36        | 37.79        |
> | 153       | 38.60        |
> | 138       | 42.86        |
> | 206       | 44.21        |
> | 234       | 46.30        |
> | 289       | 46.60        |
> | 275       | 47.89        |
> | 308       | 48.55        |

---

> > ### Author Response · Authors · 2025-11-20
> >
> > ## Regarding ablation experiments on Calibration Data
> > Please find our ablation on calibration data in **section H.1**
> >
> >
> > ## If the performance of the original model you choose is too poor, will it affect the merged or spliced pruned model?
> >
> >
> > Thank you for this important question. We need to distinguish between two scenarios:
> >
> > **Scenario 1: All candidate models are poor.** If all initial models in the candidate pool perform poorly, our method will indeed struggle to produce a high-performing pruned model. This is expected, as the searched model's capabilities cannot emerge from nothing;
> >
> > **Scenario 2: Some candidates are poor, but others are strong.** Our algorithm is **robust to the presence of poor models** in the candidate pool. Since our search algorithm considers all candidate models as starting points and selectively combines their best layers, the final pruned model will **not perform worse than the best individual candidate**. Poor candidates are naturally filtered out during the search process.
> >
> > We demonstrate this robustness through experiments with different model combinations in **Section 5.3 and Appendix H.6**, showing that our method consistently extracts the best capabilities from the candidate pool regardless of the presence of weaker models.
> >
> >
> > ## How much impact will switching to other merge methods have on the performance of the pruned model?
> >
> > Thank you for this comment. We have now also evaluated our method combined with TIES [5], an advanced sparse-based merging technique. The results below show that integrating more advanced merging techniques could bring additional performance gains.
> >
> > | Type | CMNLI | HeSW | PIQA | CHID | WSC_P | WSC_G | CSQA | BoolQ | MMLU | CMLU | Race_M | Race_H | XSum | C3 | Avg |
> > |------|-------|-------|-------|-------|-------|-------|-------|-------|-------|-------|-------|-------|-------|-------|-------|
> > | Base | 32.98 | 71.34 | 78.18 | 41.56 | 37.50 | 38.46 | 55.04 | 70.70 | 46.67 | 31.88 | 35.53 | 33.36 | 19.55 | 43.84 | 45.47 |
> > | Math | 32.99 | 68.60 | 75.79 | 39.71 | 39.42 | 36.54 | 50.78 | 69.36 | 42.04 | 32.16 | 30.36 | 36.42 | 20.88 | 43.45 | 44.25 |
> > | LM | 31.30 | 71.28 | 75.95 | 36.11 | 63.46 | 59.62 | 64.29 | 74.77 | 48.30 | 33.93 | 52.52 | 55.22 | 22.45 | 47.56 | 52.63 |
> > | Code | 32.99 | 70.27 | 78.62 | 41.61 | 36.54 | 41.35 | 57.41 | 71.04 | 46.22 | 32.20 | 41.25 | 39.69 | 18.79 | 46.25 | 46.73 |
> > | Ours（task arithmetic） | 35.46 | 54.43 | 67.74	 | 23.63 | 63.46 | 43.27 | 62.9 | 75.08 | 48.75 | 33.86 | 55.35 | 58.64 | 12.99 | 44.16 | 48.55 |
> > | Ours（ties） | 34.93 | 54.53 | 68.67 | 33.43| 62.38 | 41.38 | 60.89 | 73.46 | 46.39 | 33.65 | 52.29 | 55.97 | 20.12 | 42.63 | 48.62 |
> >
> > ## Scaling to 70B models
> >
> > We have now scaled our experiments to 70B models, pruning 30% of the layers. Due to constrained computational resources and the restricted runtime window in the past few days, we conducted 100 search iterations for now. However, despite this constrained search budget, the experimental results show that our method already achieves better performance compared to baseline approaches, as shown in the table below.
> >
> > | Dataset | CMNLI | HeSw | PIQA | WSCP | WSCG | CSQA | BoolQ | MMLU | CMMLU | Race-H | Race-M | Avg. |
> > |---------|-------|------|------|-----|-------|-------|------|-------|--------|--------|------|------|
> > | model_1 [1]  | 33.09 | 79.44 | 82.43| 36.54| 34.62 | 80.02 | 70.86| 67.48 | 53.63  | 76.30  | 79.32|63.07|
> > | model_2 [2]  | 34.52 | 78.19 | 80.14| 63.46| 61.54 | 78.21 | 82.94| 61.71  | 43.33  | 63.44  | 71.66|65.38|
> > | ShortGpt   | 32.33  | 25.59| 51.03| 63.46| 63.46 | 76.17  | 71.62| 26.96 | 24.74   | 62.69   | 71.73 |51.80|
> > | Ours(100 trials) | 31.80 | 40.80| 54.24| 61.02 | 62.48  | 78.20  | 74.30 | 35.70  | 33.60   | 60.80   | 70.32 |54.84|
> >
> >
> >
> > [1]meta-llama/Llama-2-70b
> >
> > [2]meta-llama/Llama-2-70b-chat-hf
> >
> > [3] Reassessing layer pruning in llms: New insights and methods. arXiv preprint arXiv:2411.15558
> >
> > [4] Streamlining redundant layers to compress large language models. arXiv preprint arXiv:2403.19135.
> >
> > [5]Yadav, Prateek, et al. Ties-merging: Resolving interference when merging models.
> >
> >
> >
> >
> > Thank you for your detailed comments. Please don’t hesitate to reach out with any further questions. We would be happy to discuss further.

---

> > > ### Author Response · Authors · 2025-11-26
> > >
> > > Thanks again for your feedback. We made a strong effort to address all your points. We would greatly appreciate it if you could let us know whether our responses address your concerns or if any further clarification is needed.

---

> > > > ### Comment · Reviewer_unTV · 2025-11-26
> > > >
> > > > Thank you for the author's detailed reply. Most of the problems have been resolved, therefore I will increase my score to 8.

---

### Official Review · Reviewer_665n · 2025-10-31

**Soundness:** 2
**Presentation:** 2
**Contribution:** 2
**Rating:** 4
**Confidence:** 3

**Summary:**

GPTailor is a structured pruning framework that compresses LLMs by cutting, selecting, and stitching layers from multiple fine-tuned variants. Using zero-order (Bayesian/SMAC) search, it finds layer combinations that balance accuracy and size. On LLaMA-2-13B models, GPTailor retains ~97% accuracy with ~25% fewer parameters, outperforming prior structured pruning baselines.

**Strengths:**

1. Introduces a creative, structured pruning paradigm that assembles a smaller model by reusing layers from multiple fine-tuned variants.

2. Frames pruning as a meta-optimization problem, offering a search-based alternative to rule-based or heuristic layer removal.

3. The method is conceptually simple and potentially compatible with existing LLM fine-tuning workflows.

**Weaknesses:**

1. Since GPTailor builds its pruned model by mixing layers from several fine-tuned versions, it’s hard to tell whether the reported improvements actually come from the proposed search and stitching method, or simply from combining strong models together. The paper would be more convincing if it included comparisons with random or heuristic layer combinations to separate the effect of its algorithm from that of model diversity.

2. The approach assumes that we already have multiple fine-tuned versions of the same base model lying around, which isn’t realistic in many deployment settings. Keeping and using several large checkpoints also adds heavy storage and computation overhead, making the method less appealing for practical use.

3. While GPTailor reduces parameter count, the paper doesn’t show whether this actually leads to faster inference, lower FLOPs, or any hardware-level acceleration. Without latency or wall-clock measurements, it’s difficult to judge whether the pruning translates into real performance benefits in practice.

**Questions:**

1. How do you ensure that the reported performance improvements are due to the search/stitching algorithm and not simply from  combining already-strong fine-tuned variants?

2. What is the total computational and memory overhead of the search procedure (number of model evaluations, GPU hours, etc.)?

3. Do you observe actual inference acceleration (throughput or latency) on GPU hardware after pruning?

4. How does GPTailor behave if only a single fine-tuned model is available?

---

> ### Author Response · Authors · 2025-11-20
>
> Thank you for your feedback! We appreciate that you liked the framing of pruning as a meta-optimization problem  and the novelty of our method. You can find responses addressing your questions as follows:
>
>
>
> ## Response to inference acceleration questions
>
> We want to clarify that our method performs **layer-wise (structured) pruning**, which is **inherently hardware-friendly and directly translates to real speedups without requiring specialized kernels or hardware modifications**. By removing entire layers, our approach produces a smaller dense model that runs efficiently on standard hardware using existing deep learning frameworks, ensuring immediate practical benefits in real-world deployment scenarios. The improvement in inference throughput is shown in the table below, showing approximately 42% speedup compared to the original model.
>
>
> **Measured throughput improvement:** We provide concrete inference measurements showing significant real-world speedups. For LLaMA2-13B with 25% layer pruning:
>
> | Metric | Before Pruning | After Pruning | Improvement |
> |--------|----------------|---------------|-------------|
> | **Output tokens/s** | 1,078.32 | 1,534.44 | **+42.3%** |
>
>
>
> ## Response to Performance Attribution - Why we can attribute improvements to the proposed search and stitching method
>
> We want to clarify our experimental design, which ensures that reported improvements are **attributable to our search/stitching algorithm rather than simply using strong fine-tuned variants.**
>
> We evaluate three scenarios to guarantee a clear comparison (experimental settings section 4) (results in **Tables 7 and 8**):
>
> (1) Applying each baseline pruning method individually to all candidate models and picking the strongest one,
> (2) First pruning each candidate model using the baseline method and then merging them
> (3) First merging the candidate models and then applying pruning
>
> By comparing against these three scenarios, we make sure that the benefits come specifically from our **pruning-while-merging** approach. If improvements were simply due to having strong fine-tuned variants, scenario (1) would perform comparably. If naive combination of pruned or unpruned models were sufficient, scenarios (2) and (3) would match our performance.
>
>
>
>
> ## Response to Computational and Memory Overhead of the Search Procedure
>
> Thank you for bringing this up, we agree that the computational overhead of the search is a question of direct practical relevan:
>
> **Computational cost:** The number of model evaluations is detailed in **Table 15**, and the computational cost comparison is shown in **Table 17**. Through our investigation, we find that our method achieves **better performance with lower computational cost** compared to other baselines.  (Our method requires fewer computational resources (9.35e+16  FLOPs) compared to Baseline (4.91e+19）, while achieving better performance (48.55 vs. 42.76).
>
> **Memory overhead during search:** The candidate model pool does **not need to be loaded into GPU memory simultaneously**, models can be stored on disk (RAM or SSD storage), which is inexpensive. The merging process is performed in a **parallel layer-wise manner**, meaning our **peak memory equals the size of the final pruned model**, not the sum of all candidate models. The merging itself is computationally inexpensive.
>
>
> ## How does GPTailor behave if only a single fine-tuned model is available
>
> When multiple models are unavailable, our method still improves single-model layer folding (Section 5.4). While not as effective as our proposed multi-model approach, it **still outperforms baselines with lower pruning cost** (Table 17). (Our method requires significantly fewer computational resources (1.75e+16 FLOPs) compared to Baseline (4.91e+19）, while achieving better performance (46.26 vs. 42.76).
>
>
> Thank you for your detailed comments. We hope that our additional experiments and clarifications have addressed your concerns. Please don’t hesitate to reach out with any further questions. We would be happy to discuss further.

---

> > ### Comment · Reviewer_665n · 2025-11-23
> >
> > Thank you very much for the detailed rebuttal and for providing additional experiments. After reviewing the responses carefully, I still have a few concerns that I hope the authors could comment on further, mainly related to the practical feasibility and interpretability of the reported results.
> >
> > - First, regarding the computational overhead of the search process, the provided FLOPs estimates in Tables 15 and 17 are helpful, but they do not fully address my original question. FLOPs alone are hard to translate into practical cost without knowing the actual GPU hours, wall-clock runtime, and number of forward evaluations required for the full search procedure. Since the method involves hundreds of trials in total (mentioned in Table 15), including medium and high-budget evaluations, it would be very useful to understand the concrete compute cost (e.g., total GPU hours and hardware used). This additional information would help readers assess how feasible the method is in real-world compression settings.
> >
> > - Second, about the inference acceleration, the rebuttal states that pruning 25% of layers leads to more than 40% throughput improvement. This number is surprisingly high to me, given typical characteristics of LLM inference, where throughput tends to be limited by memory bandwidth, scheduling, and KV cache behaviors rather than depth alone. Without knowing the batch size, sequence length, hardware type, and whether the measurement reflects prefill or decode throughput, it is difficult to interpret the significance of the reported speedup. Additional details on the measurement setup—especially latency metrics—would help readers better understand the practical impact of the pruning.
> >
> > I also have two additional points that I realized are important for understanding the method more clearly.
> >
> > - Since GPTailor mixes layers from multiple fine-tuned variants, it is not obvious that hidden-state distributions across different variants are aligned. Fine-tuned models often diverge in their RMSNorm statistics, activation ranges, and FFN behaviors, and layer-wise stitching could introduce representational mismatch. A brief discussion or empirical evidence showing why such cross-variant layer replacement remains stable would strengthen the paper.
> >
> > - Finally, although the authors showed that the method can work with a single model (the layer-folding setting), the main results rely on having several independently fine-tuned variants. IMHO, however, 1) storing multiple 13B checkpoints is expensive (tens to hundreds of GB), and 2) training several variants purely for pruning is unrealistic. It would be helpful if the authors could clarify how many variants are generally needed to obtain the reported gains, whether checkpoints from different fine-tuning epochs could serve as substitutes, and whether the method significantly degrades when only two or three variants are available.
> >
> > Overall, I appreciate the effort put into the rebuttal and the additional experiments. I believe clarifying the points above would make the contribution much clearer and more convincing.

---

> > > ### Author Response · Authors · 2025-11-25
> > > **Response to Reviewer 665n (1/2)**
> > >
> > > Dear Reviewer, thank you for your continued engagement with our paper. We’d be happy to provide additional details:
> > >
> > > ## Computational overhead of the search process aside from FLOPs
> > >
> > >
> > > Thank you for the question! In the original submission, we focused on describing computational costs in terms of FLOPs to remain future-proof. We have now extended this description in section H.3 of our work. In short, there are two main phases of our search process to consider for computational costs.
> > >
> > > **Computation of a new Merge**, i.e. of a new candidate point to evaluate later with the search procedure. For standard merging algorithms, such as task arithmetic, which we use in the submission, the cost of the merge is negligible, coming down to  ~3 operations per model parameter. Depending on available memory, this operation can be computed directly on the accelerator device (e.g. GPU), but this is not a requirement, as the merge can also be computed with minimal CPU RAM costs by loading parameter groups from disk, and merging. Technically, this operation can be overlapped with the next operation, but it was fast enough for us, that we did not implement this overlap. For us, it costs 11.2 secs to merge 2 7B models on GPU.
> > >
> > > **The Evaluation of the Merge**: Next, the merged point is evaluated, i.e. we measure accuracy of this checkpoint on our training task. The cost of this operation is a function of a) the size of the evaluation set and b) the type of evaluation, both of which influence the speed. However, as this step is not specific to our approach, any inference framework for fast evaluation, such as vllm can be used (as we do).
> > > For example, evaluating PIQA requires only prefilling. With vllm on our GPU V100(batch size = 16),It takes us 21.23 seconds to evaluate on 1000 samples. We also note that, due to our multi-fidelity search approach, we can often end the evaluation early, and do not need to check the full dataset (See Table 16).
> > >
> > > **Updating the coefficients of the Bayesian Hyperparameter Estimation**. We use SMAC, a well-established and optimized package for Bayesian hyperparameter optimization. As the estimation is based on random forests, it is very cheap to update. For us, one step of the update takes 2.6 seconds
> > >
> > > With our parallel acceleration strategies(e.g., simultaneous merging and evaluation), the evaluation phase becomes the dominant factor in end-to-end runtime. To provide a clearer picture, we report the evaluation wall-clock time for different datasets under various computational budgets as shown in Table Below.
> > >
> > > Overall, our current approach takes 30/35.36/60.36 seconds per round(different fidelities),  and we run 500 rounds in total.(with 41% of trials require only the smallest budget). When evaluation parallelism is disabled (parallelism = 1), the total wall-time is simply the sum of the individual evaluation times.
> > >
> > >
> > >
> > > | | Time (Seconds) |
> > > |---------|---------|
> > > | **Model(vllm) Initialization** | 19.52|
> > > | **Evaluation** | See Table Below|
> > >
> > > | Dataset | Size | Evaluation Runtime(Seconds) |
> > > |---------|------|---------------------|
> > > | **CSQA** | 100 | 2.76 |
> > > | | 300 |6.97|
> > > | | 1000 | 16.51  |
> > > | **WSC** | 100 |2.41  |
> > > | | 200 |2.50 |
> > > | | 500 |2.67 |
> > > | **PIQA** | 100 | 2.75|
> > > | | 300 | 7.00 |
> > > | | 1000 |21.23  |
> > > | **MMLU** | 100 |  2.56 |
> > > | | 300 | 6.49 |
> > > | | 1000 | 21.66  |
> > >
> > >
> > >
> > >
> > > ## Inference Acceleration
> > >
> > > Yeah, we agree that 40% is a bit faster than expected based on FLOP comparisons. This is likely caused by the specifics of vllm acceleration, likely due to being able to fit a slightly better matrix shape, due to the smaller model. We obtained this value using vLLM's benchmark with 1,000 prompts (512 input and 512 output tokens each) under default continuous batching configurations on an NVIDIA A100 GPU, reporting the output throughput in tokens/s, with the command below:
> > >
> > > ```vllm bench throughput --model Pruned_model--input-len 512 --output-len 512  --num-prompts 1000```
> > >
> > > Whereas the baseline value was computed with
> > >
> > > ```vllm bench throughput --model vanilla_model --input-len 512 --output-len 512  --num-prompts 1000```
> > >
> > > We do want to re-iterate though that how much a particular model is accelerated through layer pruning is independent of the method used for layer pruning - the same 40% speed-up would also be obtained from other methods like ShortGPT (but not the same accuracy!).
> > >
> > > [our response continues in the next message]

---

> ### Author Response · Authors · 2025-11-25
> **Response to Reviewer 665n (2/2)**
>
> ## Are hidden-state distributions across different variants aligned?
>
> We agree that finetuned models could diverge in their hidden state distributions. However, because it is unclear a priori, whether a shift in hidden states is “load-bearing”/functional, or not, this can be hard to estimate. As such, our search procedure is set up to be robust to these types of mismatches. If the search procedure repeatedly finds that a particular layer cannot be slotted into the merged model, the prior on this layer being useful decreases, and other components are used. In this way, the approach is robust to layer-wise outliers in hidden state distributions.
>
> Interestingly though, these problems seem relatively rare and, on the contrary, our analysis in Fig.5 and Fig.6 actually show that layers that are most similar to other layers, are the most likely to be pruned, whereas layers that are the most different from all others, are more likely to be included.
>
> Further, Fig.7 shows that, while your intuition is correct that the pruned models have layers that are more dissimilar (in Fig.7 “roughness” refers to the abruptness of representation changes between consecutive layers, where lower values indicate smoother feature transitions in the pruned model.), our search effectively finds layer combinations where the hidden state distributions “fit” better together, than e.g. in ShortGPT.
>
>
> ## How many variants are generally needed to obtain the reported gains?
>
> In our key experiments, as described in Table 1, we merge four model variants. For Llama-13B, which we store in bfloat16, this implies a disk requirement of 13\*2\*4=104GB of storage space, which we find very doable given modern storage device costs. An ablation with fewer models can be found in Section 5.3, where we conducted experiments by varying both the number of models and their combinations. We find that incorporating high-quality models consistently improves performance across benchmarks, while adding lower-performing models does not harm overall results (still outperforming baseline methods).
>
> We further emphasize that in our experiments, we selected language, code, and math models because they are the most readily available models for free, allowing us to better test the generalizability of our method. Higher-quality models naturally lead to greater improvements, as demonstrated in Section H.6 Table 21, where using only 2 selected models from the get-go achieves better results within the same search budget than our main paper results (which contain all four generic model variants). This suggests that even with fewer variants, performance can be maintained or improved.
>
>
> ## Can checkpoints from different fine-tuning epochs serve as substitutes?
>
>
> This is a interesting point, we tested whether checkpoints from different fine-tuning epochs could serve as substitutes. As shown below, merging checkpoints from different training steps (200k, 300k, and final) of Open-LLama-7B obtains limited gains. This aligns with prior work (e.g., SWA): merging checkpoints from the same training trajectory doesn't bring additional knowledge but rather smooths the loss landscape to improve generalization and reduce forgetting.
>
> | Model | CMNLI | HeSw | PIQA | CHID | WSCG | CSQA | BoolQ | MMLU | CMMLU | RaceH | RaceM| C3 | Ave. |
> |-------|-------|------|------|------|------|------|-------|------|-------|-------|-------|-----|------|
> |Open_Llama_step_200000[1]| 31.45 | 65.53 | 78.13 | 29.47 | 41.35 | 22.52 | 61.83 | 26.89 | 25.34 | 26.1 | 24.23 |  41.42 | 39.52 |
> |Open_Llama_step_300000[2]| 32.99 | 67.86 | 77.91 | 32.27 | 61.54 | 26.37 | 60.46 | 28.35 | 24.38 | 26.5 | 27.79 |  42.41 | 42.62 |
> |Open_Llama[3]| 30.20 | 70.26 | 79.65 | 34.87 | 63.46 | 48.57 | 63.36 | 40.44 | 28.92 | 29.65 | 29.94 |  44.49 | 46.98 |
> |ShortGPT| 32.82 | 53.91 | 67.14 | 28.07 | 43.60 |41.20 | 62.17 | 28.74 | 26.13 | 22.04 |21.80 |  39.18 | 38.90 |
> |Ours| 33.00 | 56.78 | 68.01 | 29.60 | 47.12 | 43.50 | 61.65 | 31.03 | 26.80 | 22.94 | 24.10 |  35.78 | 40.02 |
>
>
>
>
>
> [1] openlm-research/open_llama_7b_v2_step_300000
>
> [2] openlm-research/open_llama_7b_v2_step_200000
>
> [3] openlm-research/open_llama_7b_v2
>
>
> Overall, thank you again for taking the time to carefully re-read our work. We hope this clarifies your additional questions.

---

### Official Review · Reviewer_6BK6 · 2025-11-01

**Soundness:** 3
**Presentation:** 4
**Contribution:** 3
**Rating:** 8
**Confidence:** 4

**Summary:**

GPTailor attempts to compress LLMs not by pruning a single network, but rather by cutting and stitching layers taken from different fine-tuned model variants. In their technique, for every layer it can either drop it, borrow it from another candidate, or merge layers from multiple candidate models. GPTailor uses a zero-order, multi-objective search over the space to keep task performance high. On Llama-2-7B and 13B it removes about a quarter of the parameters while still keeping most of the original performance, clearly ahead of structured-pruning baselines like LLM-Pruner, SliceGPT, LaCo, and ShortGPT under similar ratios. The ablations indicate that cross-model layer merging is what really recovers performance, and that increasing the size of the candidate pool usually helps while staying stable even with weaker models. Overall, it’s a solid argument that multi-model, search-based layer tailoring is significantly more effective than single-model, metric-based pruning.

**Strengths:**

- Reframes pruning as cross-model layer selection/merging as opposed to single-model trimming, which is solid approach.
- Flexible search space (drop / pick / merge per layer).
- Strong empirical results on Llama-2-7B/13B: ~25% layers removed while retaining most of the performance performance, outperforming structured-pruning baselines by a significant margin given the same candidates.
- Useful ablations showing that merging across variants is one of the main source of quality recovery, not just cross-model picking.
- Zero-order search makes it automatable and task-agnostic if a calibration set is defined.

**Weaknesses:**

- No latency/throughput inference comparison with LLM-Pruner, SliceGPT, LaCo, or ShortGPT, so the practical speedup of the 25% layer drop is unclear.
- The search itself is heavy (500 SMAC trials, multi-fidelity), which somewhat offsets the advantages and could be hard to reproduce for bigger pools of models.
- Assumes access to task-specialized variants of the same base model, which doesn't always match deployment settings.

**Questions:**

- It would be useful to report latency/throughput to show that the ~25% layer removal/merging actually yields practical speedups over baselines like LLM-Pruner, SliceGPT, LaCo, and ShortGPT.
- How does the search cost scale in practice for bigger model pools (more than the three presented in Table 16), and is there a lighter configuration that could be used users with limited compute?

I'd be happy to reconsider my score if these questions are addressed!

---

> ### Author Response · Authors · 2025-11-20
>
> Thank you for your thorough review, and for providing insightful feedback and support. We address your questions below point by point:
>
>
> ## Response to Inference Throughput Comparison Question
>
> We need to clarify our experimental setup: we compare all methods **at the same compression ratio** (25% layer pruning), which means all baselines (LLM-Pruner, SliceGPT, LaCo, and ShortGPT) produce models with **identical layer counts after pruning** and therefore achieve **exactly the same throughput** (~42% improvement as shown below). The **differences** lie in the **pruning cost** (computational overhead during pruning) and **post-pruning accuracy** (model performance after compression). As detailed in **Table 17**, our method achieves **the best accuracy with the lowest pruning cost**.
>
> #### Throughput Improvement (LLaMA2-13B Example)
>
> For reference, we have measured the throughput gains as a function of output tokens/second for Llama-2-13B before and after pruning 25% of the layers:
>
> | Metric | Before Pruning | After Pruning | Improvement |
> |-|-|-|-|
> | **Output tokens/s** | 1,078.32 | 1,534.44 | +42.3% |
>
>
> ## Response to Model Access Questions
>
> We acknowledge that fine-tuned models may not always be readily accessible. There are two perspectives to take in response:
>
> **Single-model alternative:** When multiple models are unavailable, our method still improves single-model layer folding (Section 5.4). While not as effective as our multi-model approach, it still outperforms baselines with lower pruning cost (Table 17) (Our method requires significantly fewer computational resources (1.75e+16 FLOPs) compared to Baseline (4.91e+19）, while achieving better performance (46.26 vs. 42.76).
>
>
> **Practical motivation:** We want to further emphasize our motivation, as not all organizations can afford training large models from scratch. Fortunately, open-source models (LLaMA, DeepSeek, Qwen) have enabled domain-specific fine-tuning, resulting in thousands of models on platforms like Hugging Face. Our method offers a **new perspective**: instead of traditional single-model pruning, we learn from multiple models to create efficient compressed versions. **Given even limited target-domain examples**, any party can search existing models for optimal layer combinations. We believe that with increasing open-source model availability, **learning from multiple models is an important research direction**.

---

> > ### Author Response · Authors · 2025-11-20
> >
> > ## Response to Search Cost Concern
> >
> > Thank you for this important observation. We'd like to clarify the efficiency of our search approach and potential optimizations:
> >
> > **SMAC efficiency:** We use SMAC, a well-established state-of-the-art Bayesian optimization algorithm that combines Bayesian optimization with Hyperband. SMAC performs effective search space pruning through **successive early stopping mechanisms**, terminating unpromising configurations early in the evaluation process. The **multi-fidelity strategy** further reduces inference cost by evaluating candidates on progressively larger data subsets. we believe SMAC remains a strong and appropriate choice for our setting.
> >
> > **Further optimization potential:** We can further reduce search cost through smarter search space design and incorporating domain priors. For example, we observed that pruned layers tend to concentrate in later layers. By directly removing a portion of these layers upfront, we can significantly reduce the search space. The table below demonstrates this: using 6 models with 200 trails, we achieve better performance than what we report in the paper.
> >
> >
> > | Group         | Type     | CMNLI | HeSW | PIQA | CHID | WSC_P | WSC_G | CSQA | BoolQ | MMLU | CMLU | Race_M | Race_H | XSum | C3 | Avg |
> > |-|-|-|-|-|-|-|-|-|-|-|-|-|-|-|-|-|
> > | **Base**      | Base[1]  | 32.98 | 71.34 | 78.18 | 41.56 | 37.50 | 38.46 | 55.04 | 70.70 | 46.67 | 31.88 | 35.53 | 33.36 | 19.55 | 43.84 | 45.47 |
> > |               | Math[2]  | 32.99 | 68.60 | 75.79 | 39.71 | 39.42 | 36.54 | 50.78 | 69.36 | 42.04 | 32.16 | 30.36 | 36.42 | 20.88 | 43.45 | 44.25 |
> > |               | LM[3]    | 31.30 | 71.28 | 75.95 | 36.11 | 63.46 | 59.62 | 64.29 | 74.77 | 48.30 | 33.93 | 52.52 | 55.22 | 22.45 | 47.56 | 52.63 |
> > |               | Code[4]  | 32.99 | 70.27 | 78.62 | 41.61 | 36.54 | 41.35 | 57.41 | 71.04 | 46.22 | 32.20 | 41.25 | 39.69 | 18.79 | 46.25 | 46.73 |
> > |               | LM_CN[5] | 34.02 | 70.03 | 76.71 | 38.31 | 63.46 | 59.62 | 61.51 | 56.09 | 46.47 | 32.64 | 41.48 | 45.47 | 17.64 | 46.58 | 49.29 |
> > |               | LM_long[6] | 33.05 | 71.31 | 78.62 | 39.81 | 54.81 | 31.73 | 64.29 | 71.80 | 44.54 | 33.00 | 56.98 | 60.72 | 21.45 | 48.82 | 50.78 |
> > | **Pruned**    | ShortGPT | 34.10 | 54.18 | 64.42 | 16.83 | 61.54 | 36.54 | 55.61 | 73.21 | 36.84 | 25.61 | 42.94 | 45.89 | 10.12 | 35.73 | 42.40 |
> > |               | Ours     | 34.93 | 58.50 | 69.70 | 33.88 | 63.46 | 35.58 | 62.16 | 72.05 | 47.62 | 33.96 | 55.55 | 58.57 | 18.28 | 42.58 | 49.06 |
> >
> >
> > [1]meta-llama/Llama-2-7b
> >
> > [2]TIGER-Lab/MAmmoTH-7B
> >
> > [3]meta-llama/Llama-2-7b-chat-hf
> >
> > [4]mrm8488/llama-2-coder-7b
> >
> > [5]llama2-chinese
> >
> > [6]togethercomputer/LLaMA-2-7B-32K
> >
> >
> >
> > Thank you very much for your detailed comments, constructive suggestions and support of our submission. Please don’t hesitate to let us know if you have any further questions. We would be happy to have more discussion.

---

> ### Comment · Reviewer_6BK6 · 2025-11-26
> **Reviewer response**
>
> The authors have addressed my questions. I have increased my soundness score and recommend acceptance. Thank you.

---

### Official Review · Reviewer_Qi2B · 2025-11-01

**Soundness:** 3
**Presentation:** 3
**Contribution:** 3
**Rating:** 6
**Confidence:** 4

**Summary:**

The paper introduces a new structured pruning approach that constructs smaller LLMs by selectively removing, merging, and reusing layers from multiple fine-tuned variants of a base model. Instead of pruning a single network, it formulates pruning as a zero-order optimization problem that searches across combinations of layers from task-specialized models to maximize performance under a sparsity constraint. Experiments on the LLaMA-2 and LLaMA-3 families show that GPTailor maintains up to 97% of the original performance while removing about 25% of parameters, outperforming existing pruning methods and demonstrating strong generalization to newer model architectures

**Strengths:**

1. The idea of this paper is both interesting and innovative. By compressing and merging task-specific fine-tuned models, it achieves impressive compression performance without requiring additional fine-tuning.
2. The experiments are solid, demonstrating the method’s effectiveness across four categories and fourteen datasets.
3. The paper is clearly written and easy to follow.

**Weaknesses:**

1. Theoretical analysis is weak, and although the experimental results of the method are significant, there is a lack of theoretical explanation or visual analysis (such as layer representation similarity) on why cross-model stitching is better than single model pruning.
2. Insufficient interpretability, although a structural diagram of the pruned model is provided, there is a lack of in-depth analysis on why certain layers are retained or merged.

**Questions:**

There is a lack of in-depth analysis on why certain layers are retained or merged.

---

> ### Author Response · Authors · 2025-11-20
>
> Thank you for your constructive suggestions. We're glad that you found the idea of our work innovative, our experiments solid, and the paper easy to follow.
>
>
> To address your question, we have now updated our submission document to provide a deeper analysis (**Appendix F**) from three perspectives: the theoretical foundation of model merging, empirical observations of layer-wise patterns, and post-hoc analysis of layer characteristics.
> The new analyses show that merging consistently occurs at layers where independently fine-tuned models converge to highly similar representations, forming stable cross-model consensus. Pruning, in contrast, targets layers whose representations are redundant within each model and contribute little unique information. These two properties emerge naturally from optimization and together reveal a coherent structural principle that the method preserves layers encoding shared, essential features while removing layers with negligible representational value.
>
> We do think these additions strengthen our work and make it more interesting. Please take a look at the updated PDF to find out more.
>
> We appreciate your valuable feedback and welcome any further questions or discussions. Please let us know if there are other questions; we'd be happy to address them.

---

> > ### Comment · Reviewer_Qi2B · 2025-11-26
> >
> > Thanks for your detailed response. I still maintain my positive score.

---

### Meta-Review · Area_Chair_Y3Gs · 2026-01-06

**Summary:**

This paper introduces GPTailor, a novel structured pruning framework that compresses large language models (LLMs) by cutting, selecting, and stitching layers from multiple fine-tuned variants of a base model. By reframing pruning as a zero-order optimization problem, it achieves a superior balance between model size and performance, maintaining 97.3% of Llama2-13B's performance while removing 25% of its parameters.

In the initial review stage, the reviewers' concerns primarily focused on the actual hardware-level inference speedup, the computational overhead of the search process, and the interpretability of why cross-model stitching outperforms single-model pruning. Authors in the rebuttal provided concrete wall-clock throughput data (showing a 42.3% improvement), total GPU runtime statistics, and layer-wise similarity analyses to respond to these concerns. The majority of the concerns were successfully mitigated through additional experiments. Given the final scores (6, 8, 4, 6) and the clear statements of increasing scores by some reviewers, this paper is recommended for acceptance.

**Reviewer Concerns:**

The rebuttal successfully addressed the majority of the reviewers' technical concerns, particularly regarding practical efficiency and scalability. The authors provided concrete hardware-level measurements showing a 42.3% throughput improvement and detailed wall-clock runtime statistics for the search process, which satisfied concerns about inference speed and computational overhead. Additionally, the new experiments on 70B models and larger model pools effectively demonstrated the method's robustness across scales.

**Reviewer Scores:**

Most reviewers actively participated in the post-rebuttal discussion and provided clear, positive statements regarding the authors' clarifications. Reviewer 6BK6 explicitly increased their score from a 6 to an 8, Reviewers Qi2B and unTV maintained their positive stances.

---

### Decision · Program_Chairs · 2026-01-26

Accept (Poster)